# LEARNING-AUGMENTED ROBUST ALGORITHMIC RECOURSE

## ABSTRACT

The widespread use of machine learning models in high-stakes domains can have a major negative impact, especially on individuals who receive undesirable outcomes. Algorithmic recourse provides such individuals with suggestions of minimum-cost improvements they can make to achieve a desirable outcome in the future. However, machine learning models often get updated over time and this can cause a recourse to become invalid (i.e., not lead to the desirable outcome). The robust recourse literature aims to choose recourses less sensitive, even against adversarial model changes, but this comes at a higher cost. To overcome this obstacle, we initiate the study of algorithmic recourse through the learning-augmented framework and evaluate the extent to which a designer equipped with a prediction regarding future model changes can reduce the cost of recourse when the prediction is accurate (consistency) while also limiting the cost even when the prediction is inaccurate (robustness). We propose a novel algorithm for this problem, study the robustness-consistency trade-off, and analyze how prediction accuracy affects performance.

## 1 INTRODUCTION

Machine learning models are nowadays widely deployed even in sensitive domains such as lending or hiring. For example, financial institutions use these models to determine whether someone should receive a loan. Given the major impact that such decisions can have on people's lives, a plethora of recent work in responsible machine learning aims to make these models fair (Berk et al., 2021; Barocas et al., 2019; Zafar et al., 2017; Hardt et al., 2016), transparent (Lakkaraju et al., 2016; Rudin, 2019), and explainable (Ribeiro et al., 2016; Lundberg & Lee, 2017; Smilkov et al., 2017). A notable line of work along this direction called *algorithmic recourse* (Wachter et al., 2018; Ustun et al., 2019; Karimi et al., 2020a), provides each individual who was given an undesirable label (e.g., one whose loan request was denied) with a minimum cost improvement suggestion to achieve the desired label.

One important weakness of much of the work on algorithmic recourse is the assumption that models are fixed and do not change (Wachter et al., 2018; Ustun et al., 2019). In practice, many models are periodically updated to reflect the changes in data distribution or the environment, which can cause the recourse to become invalid, i.e., following it may not lead to a desirable outcome (Dominguez-Olmedo et al., 2022). To alleviate this problem, Upadhyay et al. (2021) proposed a recourse framework that is *robust* to adversarial changes to the model parameters and provided an algorithm called ROAR to compute robust recourses. Subsequently, Nguyen et al. (2022) proposed another algorithm (RBR for short) to improve ROAR's performance for non-linear models. While both of these works computer recourses less sensitive to adversarial model changes, this comes at the price of higher cost.

To overcome this issue, we revisit the algorithmic recourse problem through the lens of the learning-augmented framework (Mitzenmacher & Vassilvitskii, 2020) which has been used in a surge of recent work to overcome the limitations of adversarial (i.e., worst-case) analysis. Specifically, rather than assuming that the designer has no information regarding how the model can change, we assume the designer can formulate predictions regarding what these changes may be. However, crucially, these predictions are unreliable and can be arbitrarily inaccurate. Using the learning-augmented approach, our goal is to optimize the validity-cost trade-off by computing recourses that perform near-optimally when the predictions are accurate (consistency) while maintaining good performance even in the worst-case, i.e., even when the predictions are arbitrarily inaccurate (robustness).

**Our Results** In Section 3, we adapt the learning-augmented framework to algorithmic recourse, and our first result (Section 3.1) is a computationally efficient algorithm that computes a recourse with optimal robustness for generalized linear models. This is a non-convex problem and, to the best of our knowledge, this is the first optimal algorithm for any robust recourse problem. For non-linear models, we first approximate the model with a local linear model and then utilize our algorithm to provide recourse for the approximate model. In Section 4.1, we empirically study the combinations of robustness and consistency that are achievable across different datasets and models. These results indicate that the trade-off between robustness and consistency is domain-dependent and can vary greatly across different datasets and models. Furthermore, apart from the extreme measures of consistency and robustness, in Section 4.1, we also study how the quality of recourse solutions returned by our algorithm degrades as a function of the prediction error. Finally, in Section 4.2 we compare our recourses to those computed by ROAR and RBR, and we observe that our recourses have higher validity than these two baselines. Furthermore, for any fixed level of validity, our recourses generally have lower costs compared to ROAR and RBR.

## 1.1 RELATED WORK

The most closely related works are by Upadhyay et al. (2021) and Nguyen et al. (2022). We compare our approach to both papers in Section 4.2. The RoCourseNet algorithm (Guo et al., 2023) also provides robust recourse though a direct comparison with this algorithm is not possible, as it is an end-to-end approach, i.e., it simultaneously optimizes for the learned model and robust recourse while the initial model is fixed in our approach, just like in (Upadhyay et al., 2021; Nguyen et al., 2022).

The learning-augmented framework has been applied to a wide variety of settings, aiming to provide a refined understanding of the performance guarantees that are achievable beyond the worst case. One of the main application domains is the design of algorithms (e.g., online algorithms (Lykouris & Vassilvitskii, 2018; Purohit et al., 2018)), but it has also been used toward the design of data structures (Kraska et al., 2018), mechanisms interacting with strategic agents (Agrawal et al., 2022; Xu & Lu, 2022), and privacy-preserving methods for processing sensitive data (Khodak et al., 2023). This is already a vast and rapidly growing literature; see (Lindermayr & Megow, 2024) for a frequently updated and organized list of related papers. Additional related work is provided in Appendix A.

## 2 PRELIMINARIES

Consider a predictive model $f_\theta : \mathcal{X} \to \mathcal{Y}$, parameterized by $\theta \in \Theta \subseteq \mathbb{R}^d$, which maps instances (e.g., loan applicants) from a feature space $\mathcal{X} \subseteq \mathbb{R}^d$ to an outcome space $\mathcal{Y} = \{0, 1\}$. The values of 0 and 1 represent undesirable and desirable outcomes (e.g., loan denial or approval), respectively. If a model $f_{\theta_0}$ yields an undesirable outcome for some instance $x_0 \in \mathcal{X}$, i.e., $f_{\theta_0}(x_0) = 0$, the objective in recourse is to suggest the least costly way to modify $x_0$ (e.g., how an applicant should strengthen their application) so that the resulting instance $x'$ would achieve the desirable outcome under $f_{\theta_0}$. Given a cost function $c : \mathcal{X} \times \mathcal{X} \to \mathbb{R}_+$ that quantifies the cost of this transformation from $x_0$ to $x'$, the recourse is defined with the following optimization problem (Upadhyay et al., 2021):

$$\min_{x' \in \mathcal{X}} \ell\left(f_{\theta_0}(x'), 1\right) + \lambda \cdot c\left(x', x_0\right), \tag{1}$$

where $\ell : \mathbb{R} \times \mathbb{R} \to \mathbb{R}_+$ is a loss function (such as binary cross entropy or squared loss) that captures the extent to which the condition $f_{\theta_0}(x') = 1$ is violated and $\lambda \geq 0$ is a regularizer that balances the degree of violation from the desirable outcome and the cost of modifying $x_0$ to $x'$. The regularizer $\lambda$ can be decreased gradually until the desired outcome is reached. In this work, following the approach of Upadhyay et al. (2021); Rawal & Lakkaraju (2020), we assume the cost function is the $L^1$ distance i.e. $c(x, x') = \|x - x'\|_1$. This implies that all the features are manipulable, they can be changed independently, and the cost of manipulating each feature is the same.[1] We also assume the loss function $\ell$ is convex and decreasing in its first argument, which is satisfied by many commonly used loss functions such as binary-cross entropy or squared loss.

We denote the *total cost* of a given recourse $x'$ for a given model $\theta$ using

$$J(x_0, x', \theta, \lambda) = \ell\left(f_\theta(x'), 1\right) + \lambda \cdot c(x', x_0). \tag{2}$$

---

[1]Our framework can easily handle the case where the features can be modified independently but the cost of modifying each feature is different (see Section 5 for a discussion).

To further simplify notation, we ignore the dependence of $J$ on $x_0$ and $\lambda$ and write $J(x', \theta)$ instead.

The objective defined in Equation (1) assumes that the parameters of the model remain the same over time, but this does not capture the fact that, in practice, predictive models may be periodically retrained and updated (Upadhyay et al., 2021). These updates can cause a recourse that is valid in the original model (i.e., one that would lead to the desirable outcome in that model) to become invalid in the updated model (Dutta et al., 2022; Black et al., 2022). It is, therefore, natural to require a recourse solution whose validity is robust to (slight) changes in the model parameters.

In line with prior work on robust recourse (Upadhyay et al., 2021; Black et al., 2022), we assume that the parameters of the updated model can be any $\theta' \in \Theta_\alpha$, where $\Theta_\alpha \subseteq \Theta$ is a "neighborhood" around the parameters, $\theta_0$, of the original model. Specifically, this neighborhood is defined using the $L^\infty$ distance and a parameter $\alpha$, so that a. Given $\theta_0$ and $\Theta_\alpha$, the *robust* solution would be to choose a recourse $x_r$ that minimizes the total cost assuming the parameters of the updated model $\theta' \in \Theta_\alpha$ are chosen adversarially, i.e.,

$$x_r \in \arg \min_{x' \in \mathcal{X}} \max_{\theta' \in \Theta_\alpha} J(x', \theta'). \tag{3}$$

See Section 5 for a discussion regarding alternative ways of defining model change.

## 3 LEARNING-AUGMENTED FRAMEWORK FOR ROBUST RECOURSE

Choosing the robust recourse $x_r$ according to (3) optimizes the total cost against an adversarially chosen $\theta \in \Theta_\alpha$, but this total cost may be much higher than the optimal total cost in hindsight, i.e., if we knew the new model parameters, $\theta'$. This is due to the overly pessimistic assumption that the designer has no information regarding what the realized $\theta' \in \Theta_\alpha$ may be. Aiming to overcome similarly pessimistic results in a variety of other domains, a surge of recent work has used the learning-augmented framework (Mitzenmacher & Vassilvitskii, 2020) to provide a more refined and practical analysis. This framework assumes the designer is equipped with some unreliable (machine-learned) prediction and then seeks to achieve near-optimal performance whenever the prediction is accurate while simultaneously maintaining some robustness even if the prediction is arbitrarily inaccurate.

We adapt the learning-augmented framework to the algorithmic recourse problem and assume that the designer can generate (or is provided with) an unreliable prediction $\hat{\theta} \in \Theta_\alpha$ regarding the model's parameters after the model change. For example, in the loan approval setting, a prediction can be inferred by any information regarding whether the lender would be tightening or loosening its policy over time, or even conveying more information about the changes in the form of an exact prediction for the future model. If the designer trusts the accuracy of this prediction, then an optimal solution would be to choose a recourse $x_c$ that is consistent with this prediction, $\hat{\theta}$, i.e.,

$$x_c \in \arg \min_{x' \in \mathcal{X}} J(x', \hat{\theta}). \tag{4}$$

However, since this prediction is unreliable, following it blindly could lead to very poor robustness. To evaluate the performance of a recourse $x'$ based on the learning-augmented framework we use the *robustness* and *consistency* measures, defined below.

**Definition 3.1** (Robustness). Given a parameter $\alpha$, the *robustness* of a recourse $x' \in \mathcal{X}$ is

$$R(x', \alpha) = \max_{\theta' \in \Theta_\alpha} J(x', \theta') - \max_{\theta' \in \Theta_\alpha} J(x_r, \theta'), \tag{5}$$

where $x_r$ is defined in Equation (3).

The robustness measure evaluates the worst-case total cost of $x'$ against an adversarial change of the model and then compares it to the corresponding total cost of $x_r$. The robustness is always at least zero (since $x_r$ could always be chosen as the proposed recourse $x'$) and lower robustness values are more desirable. Note that Upadhyay et al. (2021) measure robustness in absolute terms, but we evaluate it relative to $x_r$ to enable a more direct comparison between robustness and consistency.

**Definition 3.2** (Consistency). Given a prediction $\hat{\theta} \in \Theta_\alpha$, the consistency of a recourse $x' \in \mathcal{X}$ is

$$C(x', \hat{\theta}) = J(x', \hat{\theta}) - J(x_c, \hat{\theta}), \tag{6}$$

where $x_c$ is defined in Equation (4).

The consistency is also always at least zero (zero consistency can be achieved simply by using $x' = x_c$, i.e., by trusting the prediction) and lower consistency values are more desirable.

Therefore, choosing $x_r$ guarantees an optimal robustness of zero, but it can lead to poor consistency, and choosing $x_c$ guarantees an optimal consistency of zero, but it can lead to poor robustness. One of our main goals in this paper is to study the achievable trade-off between robustness and consistency, and we present our experimental evaluation of this trade-off in Section 4.1.

### 3.1 COMPUTING ROBUST AND CONSISTENT RECOURSES

We next propose an algorithm to compute $x_r$, i.e., a robust recourse that provides an optimal solution to the optimization problem of Equation (3) when the function $f_\theta$ is a generalized linear model. This algorithm can also be used to compute the consistent solution $x_c$ of Equation (4), by setting $\alpha = 0$. A model is generalized linear if $f_\theta$ can be written as $f_\theta(x) := g \circ h_\theta(x)$, i.e., a composition of two functions, where $h_\theta : \mathcal{X} \to \mathbb{R}$ is a linear function mapping inputs to scores and $g : \mathbb{R} \to [0, 1]$ is a non-decreasing function mapping scores to probabilities of favorable outcome (which is the label 1 in our setting). For example, setting $g$ to be the sigmoid function will recover the logistic regression.

Note that even for generalized linear functions, the objective function in (4) is non-convex (see Appendix B). Hence, gradient-based approaches such as RObust Algorithmic Recourse (ROAR) (Upadhyay et al., 2021) or Robust Bayesian Recourse (RBR) (Nguyen et al., 2022) can only converge to a *locally* optimal recourse, as opposed to our algorithm which guarantees *globally* optimal recourse. We empirically compare the performance of our algorithm to these algorithms in Section 4.2.

We introduce a few additional notations. We use sgn to denote the function $\text{sgn}(s) = \mathbb{1}[s \geq 0]$ where $\mathbb{1}$ is the indicator function. When applied to a vector, the sgn is applied element-wise to each dimension of the vector. For an integer $n \in \mathbb{N}$, $[n] := \{1, \ldots, n\}$. We also use $e_i$ to denote a $d$-dimensional unit vector with all zeros except for $i$-th coordinate which has a value of one.

---

**ALGORITHM 1:** Optimal Robust Recourse

**Input** : $x_0, \theta_0, \ell, c, \alpha$
**Output**: $x'$

1: Initialize $x' \leftarrow x_0$
2: Initialize ACTIVE=$[d]$           ▷ Set of coordinates to update

3: **for** $i \in [d]$ **do**
4:    **if** $x_0[i] \neq 0$ **then**
5:      Initialize $\theta'[i] \leftarrow \theta_0[i] - \alpha \cdot \text{sgn}(x_0[i])$    ▷ Initialization for $\theta'$ (the worst-case model)
6:    **else**
7:      **if** $|\theta_0[i]| > \alpha$ **then**
8:        Initialize $\theta'[i] \leftarrow \theta_0[i] - \alpha \cdot \text{sgn}(\theta_0[i])$
9:      **else**
10:        ACTIVE $\leftarrow$ ACTIVE $\setminus \{i\}$    ▷ Remove the coordinate that cannot improve $J$

11: **while** ACTIVE $\neq \emptyset$ **do**
12:    $i \leftarrow \arg\max_{j \in \text{ACTIVE}} |\theta'[j]|$    ▷ Next coordinate to update
13:    $\Delta \leftarrow \arg\min_\Delta J(x' + \Delta e_i, \theta') - J(x', \theta')$    ▷ Compute the best update for the selected coordinate
14:    **if** $\text{sgn}(x'[i] + \Delta) = \text{sgn}(x'[i])$ **then**
15:      $x'[i] \leftarrow x'[i] + \Delta$    ▷ Apply the update and terminate
16:      break
17:    **else**
18:      $x'[i] \leftarrow 0$    ▷ Update the coordinate but only until it reaches 0
19:      **if** $\text{sgn}(\theta_0[i]) = \text{sgn}(\theta_0[i] + \alpha \cdot \text{sgn}(x_0[i]))$ **then**
20:        $\theta'[i] \leftarrow \theta_0[i] + \alpha \cdot \text{sgn}(x_0[i])$    ▷ Modify $\theta'$ accordingly
21:      **else**
22:        ACTIVE $\leftarrow$ ACTIVE $\setminus \{i\}$
23: **return** $x'$

---

Algorithm 1 starts by computing the worst-case model $\theta'$ for the "default" recourse of $x_0$; see Lemma B.2 for why the for-loop of Algorithm 1 (lines 3-10) achieves that. Then, facing $\theta'$, the algorithm greedily modifies $x_0$ into $x_r$ while simultaneously updating $\theta'$ to ensure that it remains the worst-case model for the current recourse $x'$. In each iteration of the while loop, the algorithm

identifies the dimension $i$ of $\theta'$ that has the largest absolute value (line 12) and then computes the optimal change of $x'[i]$ if we were to keep $\theta'$ fixed (line 13). If this change does not cause $x'[i]$ to flip its sign, the adversary does indeed remain fixed, so $x'$ is the optimal recourse and the algorithm terminates. On the other hand, if the recommended change would flip the sign of $x'[i]$, this would cause the adversarial response to change as well (see Lemma B.2). The algorithm instead applies this change all the way up to $x'[i] = 0$, it updates $\theta'$ accordingly, and repeats. If, during this process, for some dimension $i$ we have $x'[i] = 0$ and an update of the adversarial model $\theta'[i]$ could cause its sign to flip, then no further change in this dimension is allowed ($i$ is removed from the ACTIVE set).

We next describe the optimality guarantee of Algorithm 1. We defer the full proof to Appendix B.

**Theorem 3.3.** *If $f_\theta$ is a generalized linear model, then Algorithm 1 returns a robust recourse $x' \in \arg\min_{x \in \mathcal{X}} \ \max_{\theta' \in \Theta_\alpha} J(x, \theta')$ in polynomial time.*

While Algorithm 1 is designed for generalized linear models, it can be extended to non-linear models by first approximating $f_\theta$ locally. This idea has also been used in prior work (Upadhyay et al., 2021; Ustun et al., 2019; Rawal & Lakkaraju, 2020). See Section 4 for more details. Moreover, in many settings, there are constraints on the space of feasible recourses (e.g., the recourse cannot decrease *age* if it is a feature), or the data contains categorical features. While Algorithm 1 cannot handle such cases, similar to prior work (Upadhyay et al., 2021; Nguyen et al., 2022; Guo et al., 2023), the recourse of Algorithm 1 can be post-processed (e.g., by projection) to guarantee feasibility.

## 4 EXPERIMENTS

In this section, we provide experimental results with real and synthetic datasets. We first describe the datasets and implementation details, and then present our findings in Sections 4.1 and 4.2.

**Datasets**   We experiment on both synthetic and real-world data. For the synthetic dataset, we follow a process similar to Upadhyay et al. (2021). We generate 1000 data points in two dimensions. For each data point, we first sample a label $y$ uniformly at random from $\mathcal{Y} = \{0, 1\}$. We then sample the instance corresponding to this label from a Gaussian distribution $\mathcal{N}(\mu_y, \Sigma_y)$. We set $\mu_0 = [-2, -2]$, $\mu_1 = [+2, +2]$, and $\Sigma_0 = \Sigma_1 = 0.5\mathbb{I}$ (see Figure 1(a) in (Upadhyay et al., 2021)). We also use two real datasets. The first dataset is the German Credit dataset (Hofmann, 1994) which consists of 1000 data points each with 7 features containing information about a loan applicant (such as age, marital status, income, and credit duration), and binary labels good (1) or bad (0) determines the creditworthiness. The second dataset is the Small Business Administration dataset (Min Li & Taylor, 2018) which contains the small business loans approved by the State of California from 1989 to 2004. The dataset includes 1159 data points each with 28 features containing information about the business (such as business category, zip code, and number of jobs created by the business) and the binary labels indicate whether the small business has defaulted on the loan (0) or not (1). For real-world data, we normalize the features. We use the datasets to learn the initial model $\theta_0$. Experimental result for a larger dataset is reported in Appendix C.4.

**Implementation Details**   We use 5-fold cross-validation in our experiments. We use 4 folds from the data to train the initial model $\theta_0$ and use the remaining fold to compute the recourse. The recourse is only computed for instances that receive an undesirable label (0) under $\theta_0$. We report average values (over folds and test instances) in all our experiments. We used logistic regression as our linear model and trained it using Scikit-Learn. As our non-linear model, we used a 3-level neural network with 50, 100, and 200 nodes in each successive layer. The neural network uses ReLU activation functions, binary cross-entropy loss, and Adam optimizer, and is trained for 100 epochs using PyTorch. The selected architecture is identical to prior work (Upadhyay et al., 2021). See Appendix C.1.

To generate robust or consistent recourses for linear models we implemented Algorithm 1. We used the code from (Upadhyay et al., 2021) and (Nguyen et al., 2022) for ROAR and RBR's implementation as our baselines. Similar to (Upadhyay et al., 2021), when the model is non-linear we first approximate it locally with LIME (Ribeiro et al., 2016) and use the local model to generate recourse with either Algorithm 1 or ROAR, resulting in potentially different parameters $\theta_0$ for different instances.

We next describe our choices for parameters for Algorithm 1 and ROAR. We use binary cross-entropy as the loss function $\ell$ and $L^1$ distance as the cost function $c$. We use $L^\infty$ norm to measure the closeness

in the space of model parameters. In Section 4.1, we follow a similar procedure as in (Upadhyay et al., 2021) for selecting $\alpha$ and $\lambda$. We fix an $\alpha$ and greedily search for $\lambda$ that maximizes the recourse validity under the original model $\theta_0$. We study the effect of varying $\alpha$ and $\lambda$ in Section 4.2 and Appendix C.3. In each experiment, we specify how $\hat{\theta}$ is selected. See our code and Appendix C.1.

## 4.1 FINDINGS: LEARNING-AUGMENTED SETTING

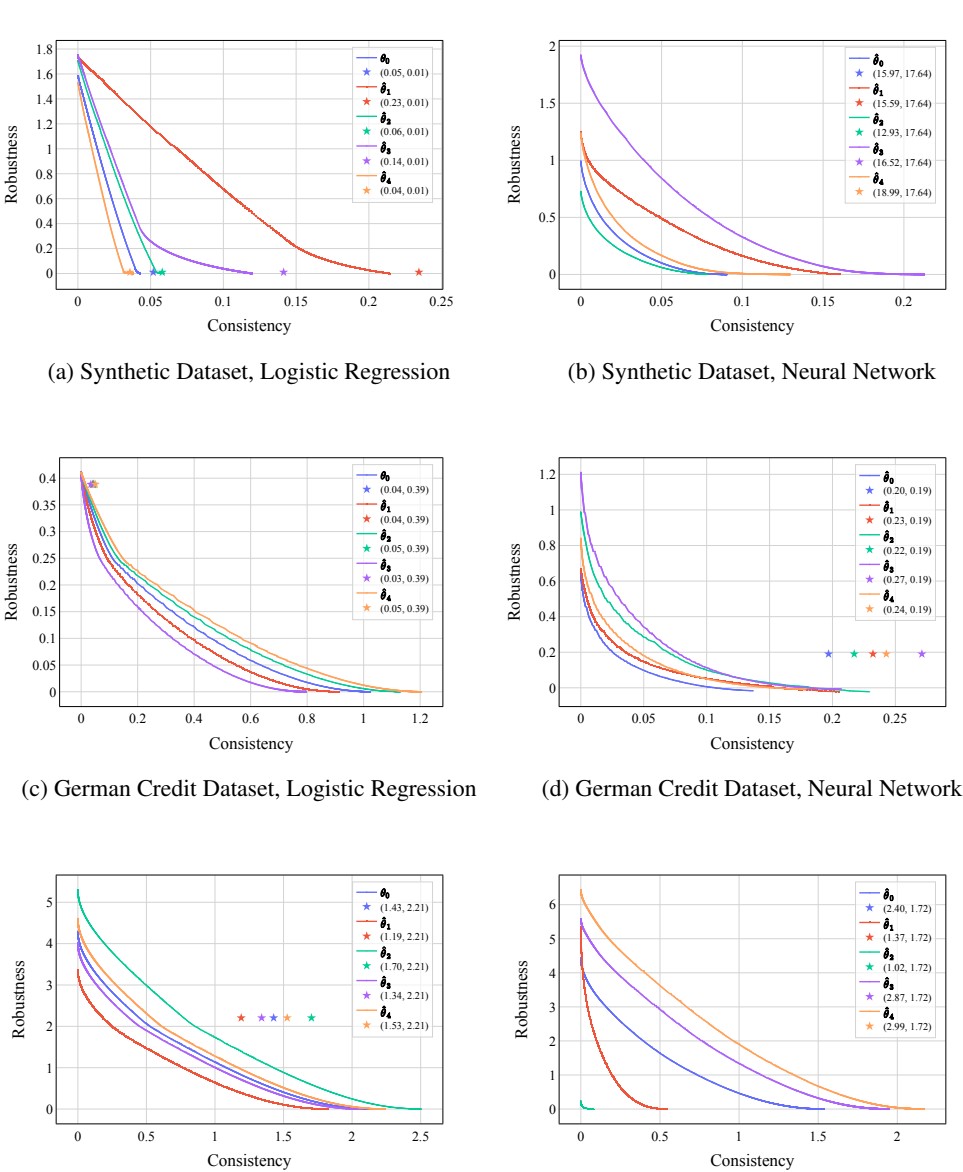

(a) Synthetic Dataset, Logistic Regression

(b) Synthetic Dataset, Neural Network

(c) German Credit Dataset, Logistic Regression

(d) German Credit Dataset, Neural Network

(e) Small Business Dataset, Logistic Regression

(f) Small Business Dataset, Neural Network

Figure 1: The Pareto frontier of the trade-off between robustness and consistency for $\alpha = 0.5$: logistic regression (left) and neural network (right). Rows correspond to datasets: synthetic (top), German (middle), and Small Business (bottom). In each subfigure, each curve shows the trade-off for different predictions. The robustness and consistency of ROAR solutions are mentioned in parentheses and depicted by stars. Missing stars are outside of the range of the coordinates of the figure.

**Robustness-Consistency Trade-off** To study the trade-off between consistency and robustness, we generated 5 predictions. For logistic regression models, we generated 4 perturbations of the original model by adding or subtracting $\alpha$ in each dimension. For neural network models, we added the perturbation to the LIME approximation of the initial model $\theta_0$. Along with $\theta_0$, these form the 5 model parameters we used as predictions. We use $\alpha = 0.5$ for the trade-off results (see Appendix C.3 for different values of $\alpha$). To compute the trade-off, for each given prediction $\hat{\theta}$, we solve for $\arg\min_{x'} \max_{\theta' \in \Theta_\alpha} \beta \cdot J(x', \theta') + (1 - \beta) \cdot J(x', \hat{\theta})$ for varying $\beta \in [0, 1]$. To solve this optimization problem, we used a variant of Algorithm 1 (see Appendix C.2). Once, we compute the solution to this optimization problem, we can compute the robustness and consistency of the solution using Equations (5) and (6).

In Figure 1, each row corresponds to a different dataset: the synthetic dataset on the top row, the German dataset on the middle row, and the Small Business dataset on the bottom row. In the left panel, the initial model $f_{\theta_0}$ is a logistic regression while, in the right panel, the initial model $f_{\theta_0}$ is a 3-layer neural network. In each sub-figure of Figure 1, each curve shows the Pareto frontier of the trade-off between the robustness and consistency of recourses by Algorithm 1 for different predictions (indicated by different colors). The bottom right point of each curve corresponds to $\beta = 1$ i.e., the optimal robust recourse $x_r$ which has a robustness of 0 due to optimality of Algorithm 1 but might have different consistency depending on the prediction. Similarly, the top left point of each curve corresponds to $\beta = 0$ i.e., the optimal consistent recourse with a consistency of 0. These solutions might have different robustness depending on the distance between the prediction and the worst-case model for robustness. In each subfigure, we use stars to show the robustness and consistency of the recourse provided by ROAR for each prediction. Since sometimes the stars fall outside of the coordinates of the figure we also specify the consistency and robustness of the ROAR recourse as a pair of numbers in the legend (see e.g., Figures 1b and 1f).

We observe that the sub-optimality of ROAR in terms of robustness compared to our optimal algorithm can vary greatly across different datasets and models. For example, while Figure 1a shows that the robustness of ROAR is only 0.01 higher than our algorithm for logistic regression models trained on the synthetic dataset, the amount of sub-optimality increases significantly to 17.64 on neural network models on the same dataset and, hence, is not displayed in Figure 1b.

**Smoothness** We also study how errors in the prediction can affect the quality of the recourse solution. In particular, for each dataset and model pair, we compute a *correct prediction* regarding the future model following the approach of (Upadhyay et al., 2021): this future model is derived by either shifting the data or as the result of temporal changes in data collection (see Appendix C.1 for more details). We then create additional, incorrect, predictions by adding perturbations to each coordinate of the correct prediction. We use four different values for the amount of added perturbations: $\{+\epsilon, -\epsilon, +2\epsilon, -2\epsilon\}$. The difference in the magnitude of perturbations allows us to generate predictions with different distances from the correct prediction: The $\epsilon$ values depend on both the dataset and the trained model and are chosen to ensure that all the predictions are within $\alpha = 1$ distance of the original model. See Appendix C.1 for details.

Given a prediction $\hat{\theta}$, a learner can utilize this prediction to generate a recourse. We assume the learner solves the optimization $\arg\min_{x'} \max_{\theta' \in \Theta_\alpha} \beta \cdot J(x', \theta') + (1 - \beta) \cdot J(x', \hat{\theta})$ to generate a recourse. By varying $\beta$ from 0 to 1, we can simulate a diverse set of strategies for the learner: $\beta = 1$ corresponds to a learner that ignores the prediction and returns the robust recourse, $\beta = 0$ corresponds to a learner that fully trusts the prediction and returns the consistent solution and $\beta \in (0, 1)$ simulate learners which lie in between the two extremes.

To measure the performance as a function of the prediction error, i.e., the *smoothness*, we use $J(x'(\beta, \hat{\theta}), \hat{\theta}_*) - J(x'(\hat{\theta}_*), \hat{\theta}_*)$ where $x'(\beta, \hat{\theta})$ is the recourse returned by the learner given prediction $\hat{\theta}$ and using parameter $\beta$, $\hat{\theta}_*$ is the correct prediction and $x'(\hat{\theta}_*)$ is the consistent recourse for the correct prediction. The smoothness is non-negative and it is 0 if the learner is provided with the correct prediction ($\hat{\theta} = \hat{\theta}_*$) and fully trusts the prediction ($\beta = 0$) to compute its recourse.

The results are summarized in Figure 2. Each row corresponds to a different dataset: synthetic (top row), German (middle row), and Small Business Administration (bottom row). The left panel shows the results for logistic regression models while the right panel corresponds to neural network models. In each sub-figure of Figure 2, each curve shows the smoothness of the learner for a given prediction

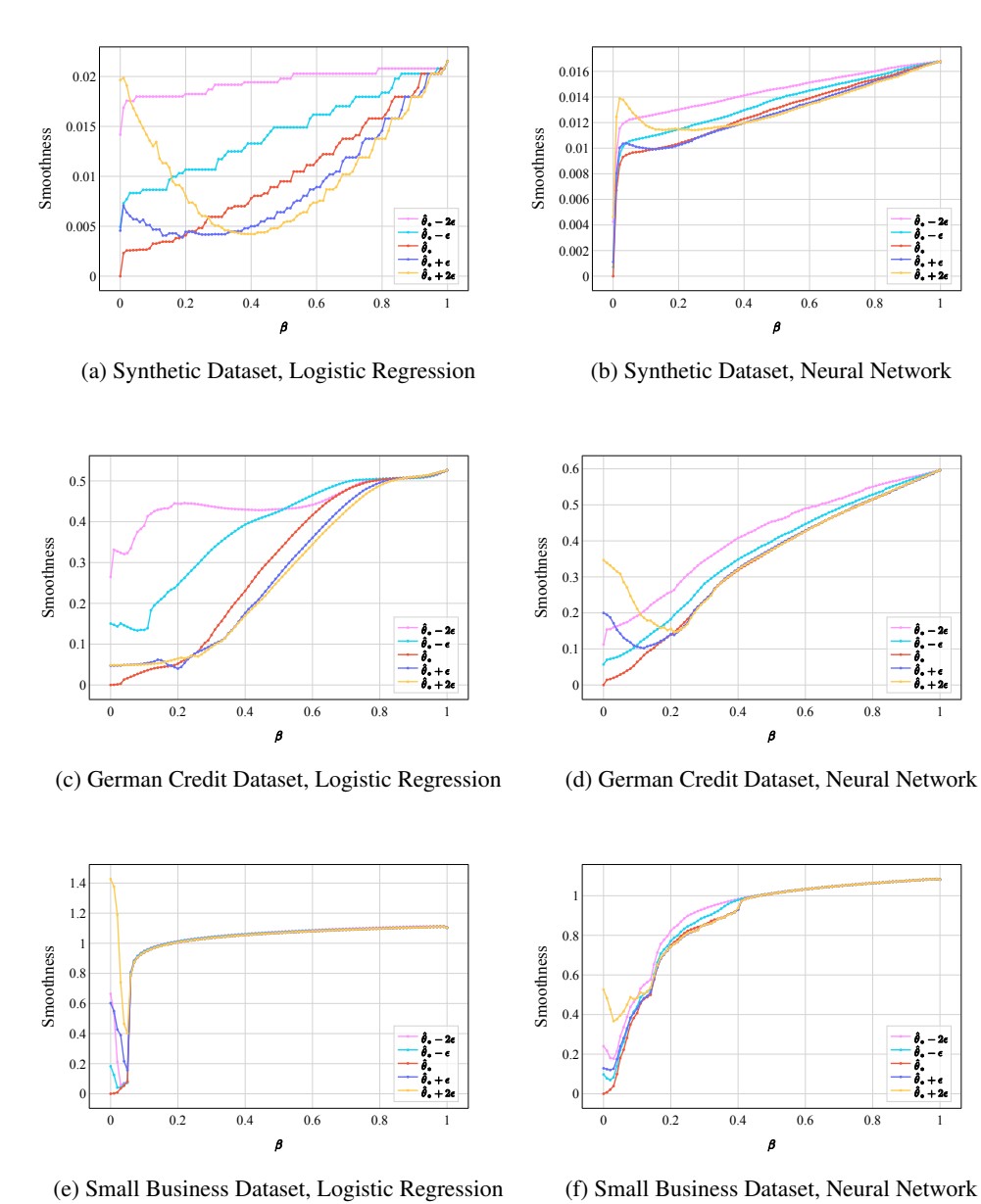

(a) Synthetic Dataset, Logistic Regression

(b) Synthetic Dataset, Neural Network

(c) German Credit Dataset, Logistic Regression

(d) German Credit Dataset, Neural Network

(e) Small Business Dataset, Logistic Regression

(f) Small Business Dataset, Neural Network

Figure 2: The smoothness analysis of recourse solutions for predictions with different accuracies: logistic regression (left panel) and neural network (right panel). Rows correspond to datasets: synthetic (top), and Small Business (bottom). In each subfigure, each curve corresponds to a different prediction and tracks the total cost of the learner as a function of $\beta$ for the given prediction.

as a function of $\beta$. There are 5 lines in each subfigure: one for the correct prediction (denoted as $\hat{\theta}_*$) and each for the four perturbations (denoted as $\hat{\theta}_* \pm$ the perturbation).

If we focus on $\beta = 0$, we observe that the cost does, in general, increase as a function of the prediction "error" (its distance from the true model parameters: either $\hat{\theta}_* + 2\epsilon$ or $\hat{\theta}_* - 2\epsilon$). However, as $\beta$ increases, the total cost of recourses for different predictions converges to the same value since at $\beta = 1$ the learner increasingly ignores the prediction. In some cases, this convergence occurs at smaller values of $\beta$ (e.g., Figure 2e) but other cases require $\beta$ to be very close to 1 (e.g., Figure 2a). Finally, while the total cost monotonically increases as $\beta$ increases when using the correct prediction,

using incorrect predictions can result in interesting non-monotone behavior and lead to recourses that have better performance compared to using the correct prediction (e.g., Figure 2a).

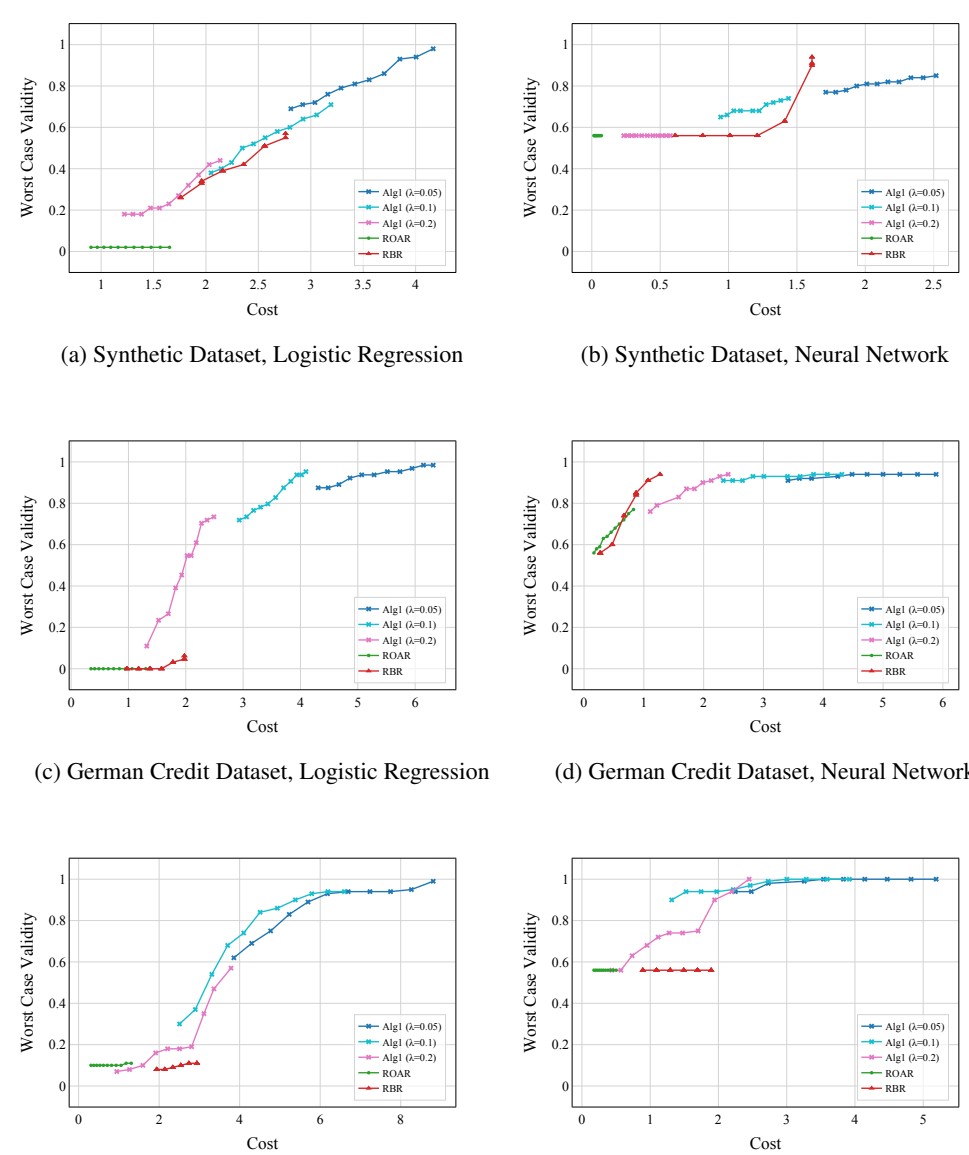

Figure 3: The Pareto frontier of the trade-off between the worst-case validity and the cost of recourse solutions. The left panel is for the logistic regression while the right panel is for a 3-layer neural network. Rows correspond to datasets: synthetic (top), and Small Business (bottom). In each subfigure, each curve shows the trade-off for different methods as mentioned in the legend.

## 4.2 FINDINGS: COMPARISON WITH ROAR AND RBR

In Section 4.1, we showed that Algorithm 1 computes recourses with significantly smaller robustness costs compared to ROAR. In this section, we perform a more detailed comparison of our algorithm with ROAR and RBR (Nguyen et al., 2022) which are two main prior approaches to computing robust recourse. We first compute the robust recourse with our algorithm and these baselines. To perform a similar comparison as in prior work, we then break down the total robustness to understand the effect

of each of the terms in Equation 1. The first term, $\ell\left(f_{\theta'}(x'), 1\right)$, is a proxy for *worst-case validity* and the second term, $c\left(x', x_0\right)$, is the cost of modifying $x_0$.

More formally, focusing on instances with undesirable labels under the original model $f_{\theta_0}$, worst-case validity is defined as the fraction of these instances labeled with the desirable label post recourse. The labels of these instances are determined using the worst-case model within $\alpha$ distance of $f_{\theta_0}$. This worst-case model is the one that minimizes the fraction of instances that achieve the desirable label post-recourse. We highlight that as opposed to computing a possibly different worst-case model for each instance, as is done in Sections 3 and 4.1, we compute a single worst-case model to be consistent with how validity is defined in prior work (Upadhyay et al., 2021; Nguyen et al., 2022). We use projected gradient ascent to compute this worst-case model. See Appendix C.1.

Figure 3 depicts the Pareto frontier of the trade-off between the worst-case validity and cost of recourse for all datasets (rows) and models (columns). The Pareto frontier for RBR is obtained by varying the parameters of RBR exactly as is done in (Nguyen et al., 2022). In particular, we set the ambiguity sizes $\epsilon_1$ and $\epsilon_0$ to $\epsilon_0, \epsilon_1 \in [0, 1]$ with increments of 0.5, and the maximum recourse cost $\delta = \|x_0 - x_r\|_1 + \delta_+$ to $\delta_+ \in [0, 1]$ with increments of 0.2. The Pareto frontier for Algorithm 1 and ROAR is computed by varying $\alpha \in [0.02, 0.2]$ in increments of 0.02. For ROAR and our algorithm, we used three different $\lambda$s: 0.05, 0.1, and 0.2, and the trade-off for each choice is plotted with a different color. To avoid overcrowding, we only included the results of $\lambda = 0.05$ for ROAR. Increasing $\lambda$ to 0.1 does not change the trade-off and $\lambda = 0.2$ degrades the validity even further.

For logistic regression models (left panel in Figure 3), where the *optimal* worst-case model can be computed efficiently, our algorithm (regardless of choice of $\lambda$) *almost* always dominates RBR and ROAR in terms of both cost and worst-case validity. The validity of RBR and ROAR remains low over all datasets specifically in real-world datasets. On the other hand, our algorithm displays a wide range for validity: low for $\lambda = 0.2$ to almost 1 for $\lambda = 0.05$. Consistent with prior work (Pawelczyk et al., 2023a), the cost of the recourse increases significantly for validity values that reach 1.

For neural network models (right panel in Figure 3), gradient ascent is not guaranteed to find the *optimal* worst-case model. Perhaps due to this, we observe that the worst-case validity of RBR and ROAR are improved for neural network models compared to the logistic regression models. However, the validity is still generally lower compared to ours (except for Figure 3b). The worst-case validity of our algorithm for neural network models improves for high $\lambda$s but degrades slightly for low $\lambda$s, compared to logistic models. However, the cost of recourse is generally lower for all algorithms perhaps again due to the challenge of computing the optimal worst-case model (Guo et al., 2023).

## 5 CONCLUSION AND DISCUSSION

We initiated the study of the algorithmic recourse problem through the learning-augmented framework. One limitation of our work is the assumption that the cost of modifying features is the same for all inputs and measured using the $L^1$ norm. While our framework can handle customizable weights for different inputs, using any norm as the cost function implies that the features can be modified independently and does not consider the causal relationship and dependencies between different features. Some prior work on algorithmic recourse aims to understand the *actionability* of the recourse solution as well as considering these casual relationships (Karimi et al., 2020a; Joshi et al., 2019; Pawelczyk et al., 2020). We leave the study of these issues as future work (see Appendix A).

Our notion of robustness and consistency measures the performance of the algorithm against the optimal robust or consistent solution in an additive manner (similar to *regret* in online learning (Cesa-Bianchi & Lugosi, 2006)). This comparison can also be done multiplicatively, similar to the *competitive ratio* for online algorithms (Mitzenmacher & Vassilvitskii, 2020). We leave the computation of robust and consistent recourses under a multiplicative comparison benchmark as well as the study of their trade-off as future work. Moreover, studying a weaker notions of model change such as measuring the model change by $L^1$ or $L^2$ norm (Upadhyay et al., 2021) or studying alternative ways of formalizing model change (see e.g., (Hamman et al., 2023)) is an interesting direction for future work. Finally, we assumed the prediction about the updated model is explicitly given. In practice, the feedback about the updated model might be "weaker" or even noisy (Bechavod et al., 2022). Incorporating such feedback into our framework is an exciting future work direction.

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

## A  ADDITIONAL RELATED WORK

The emerging literature on the interpretability and explainability of machine learning systems mainly advocates for two main approaches. The first approach aims to build inherently simple or interpretable models such as decision lists (Lakkaraju et al., 2016) or generalized additive models (Wang et al., 2017; Yang et al., 2017). These approaches provide *global* explanations for the deployed models. The second approach attempts to explain the decisions of complex black-box models (such as deep neural networks) only on specific inputs (Ribeiro et al., 2016; Lundberg & Lee, 2017; Smilkov et al., 2017; Sundararajan et al., 2017; Selvaraju et al., 2017; Agarwal et al., 2021; Koh & Liang, 2017; Barshan et al., 2020; Ehyaei et al., 2024). These approaches provide a *local* explanation of the model and are sometimes referred to as post-hoc explanations.

Recourse is a post-hoc counterfactual explanation that aims to provide the lowest cost modification that changes the prediction for a given input with an undesirable prediction under the current model (Wachter et al., 2018; Ustun et al., 2019; Looveren & Klaise, 2021; Rawal & Lakkaraju, 2020; Slack et al., 2021; Pawelczyk et al., 2023a). Since its introduction, different formulations have been used to model the optimization problem in recourse (see (Verma et al., 2020) for an overview). Wachter et al. (2018) and Pawelczyk et al. (2023a) considered score-based classifiers and defined modifications to help instances achieve the desired scores. On the other hand, for binary classifiers, Ustun et al. (2019) required the modification to result in the desired label. Roughly speaking, the first setting can be viewed as a relaxation of the second setting and we follow the second formulation in our work.

The follow-up works on the problem study several other aspects such as focusing on specific models such as linear models (Ustun et al., 2019) or decision-trees (Kanamori et al., 2024; Bewley et al., 2024), understanding the setting and its implicit assumptions and implications (Barocas et al., 2020; Venkatasubramanian & Alfano, 2020; Fokkema et al., 2024; Gao & Lakkaraju, 2023), attainability or actionability (Joshi et al., 2019; Pawelczyk et al., 2020; Karimi et al., 2020a; Ustun et al., 2019), imperfect causal knowledge (Karimi et al., 2020b), fairness in terms of cost of implementation for different subgroups (Guldogan et al., 2023; Gupta et al., 2019; Heidari et al., 2019) and repeated dynamics (Fonseca et al., 2023; Bell et al., 2024; Ehyaei et al., 2024). Extending our work to account for these different aspects of recourse is left for future work.

In addition to the closest related work mentioned in Section 1.1, Pawelczyk et al. (2023b) studies how model updates due to the "right to be forgotten" can affect recourse validity. Dutta et al. (2022) studied the robustness in recourse for tree-based ensembles. Dominguez-Olmedo et al. (2022) showed that minimum cost recourse solutions are provably not robust to adversarial perturbations in the model and then present robust recourse solutions for linear and differentiable models. Black et al. (2022) observe that recourse in deep models can be invalid by small perturbations and suggest that the model's Lipschitzness at the counterfactual point is the key to preserving validity. Very recently, Hamman et al. (2023) proposed a new notion of model change which they coin "naturally-occurring" model change and provide recourse with theoretical guarantees on the validity of the recourse.

The literature on recourse is also closely related to the vast body of work on robust machine learning (Madry et al., 2018; Wong & Kolter, 2018; Athalye et al., 2018). Pawelczyk et al. (2022) studied the connections between various recourse formulations and their analogs in the robust machine-learning literature.

## B   OMITTED DETAILS FROM SECTION 3.1

**Non-convexity of the Optimization Problem in Equation 3 for Linear Models**   We provide a concrete example that makes it easy to verify the non-convexity of the optimization problem in Equation 3 even for linear models. Consider an instance in one dimension where $x_0 = [1, 1]$ (note that the second dimension is the unchangeable intercept), $\theta_0 = [0, 0]$, $\ell$ is squared loss, $\alpha = 0.5$, and $\lambda = 1$. For any recourse, $x_r = [x, 1]$ (note that the intercept cannot change), the worst-case $\theta'$ is of the form $[0.5\text{sign}(x), -0.5]$ since $\alpha$ is 0.5 and $\theta_0$ is 0 in both dimensions. The cost of recourse for $x_r$ can be written as $1/\left(e^{0.5x\text{sign}(x)-0.5}\right)^2 + |x - 1|$. This function is not convex.

**Proof of Theorem 3.3**   To prove Theorem 3.3 and verify the optimality of Algorithm 1, we first make some observations and prove some useful lemmas. Without loss of generality, throughout this section we will be assuming that $\theta_0[i] \neq \theta_0[j]$ for any two dimensions $i \neq j$.[2]

**Observation B.1.**   For a fixed set of parameter values, the problem of optimizing robustness in our setting can be captured as computing a recourse $x_r$ aiming to minimize the value of a function $J(\cdot)$ whose value depends only on the distance cost of $x'$, i.e., $\|x' - x_0\|_1$, and its inner product with an adversarially chosen $\theta' \in \Theta_\alpha$. Formally, our goal is to compute a recourse $x_r$ such that:

$$x_r \in \arg\min_{x' \in \mathcal{X}} \max_{\theta' \in \Theta_\alpha} J(\|x' - x_0\|_1, \ x' \cdot \theta').$$

Also, $J(\cdot)$ is a linear increasing function of $\|x' - x_0\|_1$ and a convex decreasing function of $x' \cdot \theta'$.

Observation B.1 provides an alternative interpretation of the problem: by choosing a recourse $x'$, we suffer a cost $\|x' - x_0\|_1$ and the adversary then chooses a $\theta'$ aiming to minimize the value of the inner product $x' \cdot \theta'$. This implies that for a given $x'$ a choice of $\theta'$ is not optimal for the adversary unless it minimizes this inner product. Also, it implies that among all choices of $x'$ with the same cost $\|x' - x_0\|_1$, the optimal one has to maximize the inner product $x' \cdot \theta'$ with the adversarially chosen $\theta'$. We use this fact to prove that a recourse $x'$ is not a robust choice by providing an alternative recourse with the same cost and a greater dot product.

Our first lemma provides additional structure regarding the optimal adversarial choice in response to any given recourse $x$.

---

[2]This can be easily guaranteed by an arbitrarily small perturbation of these values without having any non-trivial impact on the model, but all of our results hold even without this assumption; it would just introduce some requirement for tie-breaking that would make the arguments slightly more tedious.

**Lemma B.2.** *For any recourse $x$, the adversarial response $\theta' = \arg\max_{\theta \in \Theta_\alpha} J(x, \theta)$ is such that $\theta'[i] = \theta_0[i] + \alpha$ for each dimension $i$ such that $x[i] < 0$ and $\theta'[i] = \theta_0[i] - \alpha$ for each dimension $i$ such that $x[i] > 0$. For any dimension $i$ with $x[i] = 0$ we can without loss of generality assume that $\theta'[i] \in \{|\theta_0[i] + \alpha|, |\theta_0[i] - \alpha|\}$.*

*Proof.* For any dimension $i$ with $x[i] = 0$, it is easy to verify that no matter what the value of $\theta'[i]$ is, the contribution of $x[i] \cdot \theta'[i]$ to the inner product $x \cdot \theta'$ is zero, so we can indeed without loss of generality assume that $\theta'[i] \in \{|\theta_0[i] + \alpha|, |\theta_0[i] - \alpha|\}$. Now, assume that $x[i] < 0$, yet $\theta'[i] < \theta_0[i] + \alpha$, and consider an alternative response $\theta''$ such that $\theta''[i] = \theta_0[i] + \alpha$ and $\theta''[j] = \theta'[j]$ for all other dimensions $j \neq i$. Clearly, $\theta'' \in \Theta_\alpha$, since $|\theta''[i] - \theta_0[i]| = \alpha$ and $|\theta''[j] - \theta_0[j]| \leq \alpha$ for all other dimensions $j \neq i$ as well, by the fact that $\theta' \in \Theta_\alpha$. Therefore, it suffices to prove that $x \cdot \theta'' < x \cdot \theta'$, as this would contradict the fact that $\theta' = \arg\max_{\theta \in \Theta_\alpha} J(x, \theta)$. To verify that this is indeed the case, note that

$$
\begin{aligned}
x \cdot \theta' - x \cdot \theta'' &= x[i] \cdot \theta'[i] - x[i] \cdot \theta''[i] \\
&= x[i] \cdot (\theta'[i] - \theta''[i]) \\
&> 0,
\end{aligned}
$$

where the first equation use the fact that $\theta'$ and $\theta''$ are identical for all dimensions except $i$ and the inequality uses the fact that $x[i] < 0$ and $\theta'[i] < \theta''[i]$. A symmetric argument can be used to also show that $\theta'[i] = \theta_0[i] - \alpha$ for each dimension $i$ such that $x[i] > 0$. $\square$

Lemma B.2 shows that for any recourse $x$, an adversarial response that minimizes $x \cdot \theta'$ is $\theta' = \theta_0 - \alpha \cdot \text{sgn}(x)$. Our next lemma shows how the adversarial response to the initial point $x_0$, (i.e., $\theta_0 - \alpha \cdot \text{sgn}(x_0)$) determines the direction toward which each dimension of $x_0$ should be changed (if at all).

**Lemma B.3.** *For any optimal recourse $x_r \in \arg\min_{x' \in \mathcal{X}} \max_{\theta' \in \Theta_\alpha} J(x', \theta')$ and every coordinate $i$, it must be that $x_r$ raises the value of the $i$-th dimension only if the adversary's best response to its original value is positive, and it lowers it only if the adversary's best response to its original value is negative. Using Lemma B.2, we can formally define this as:*

$$
\begin{aligned}
x_r[i] > x_0[i] \quad &only\ if \quad \theta_0[i] - \alpha \cdot \text{sgn}(x_0[i]) > 0 \\
x_r[i] < x_0[i] \quad &only\ if \quad \theta_0[i] - \alpha \cdot \text{sgn}(x_0[i]) < 0.
\end{aligned}
$$

*Proof.* Assume that for some dimension $i$ we have $x_r[i] > x_0[i]$ even though $\theta_0[i] - \alpha \cdot \text{sgn}(x_0[i]) < 0$, and let $x'$ be the recourse such that $x'[i] = x_0[i]$ while $x'[j] = x_r[j]$ for all other coordinates, $j \neq i$. If $\theta^*$ is the adversary's best response to $x_r$ and $\theta'$ is the adversary's best response to $x'$, then the difference between the inner product of $x' \cdot \theta'$ and $x_r \cdot \theta^*$ is:

$$
\begin{aligned}
x' \cdot \theta' - x_r \cdot \theta^* &= x'[i] \cdot \theta'[i] - x_r[i] \cdot \theta^*[i] \\
&= x_0[i] \cdot (\theta_0[i] - \alpha \cdot \text{sgn}(x_0[i])) - x_r[i] \cdot \theta^*[i] \\
&\geq x_0[i] \cdot (\theta_0[i] - \alpha \cdot \text{sgn}(x_0[i])) - x_r[i] \cdot (\theta_0[i] - \alpha \cdot \text{sgn}(x_0[i])) \\
&= (x_0[i] - x_r[i]) \cdot (\theta_0[i] - \alpha \cdot \text{sgn}(x_0[i])) \\
&> 0,
\end{aligned}
$$

where the first equation uses the fact that $x'[j] = x_r[j]$ for all $j \neq i$, the second equation uses the fact that $x'[i] = x_0[i]$ and the fact that the adversary's best response to $x_0[i]$ is $\theta_0[i] - \alpha \cdot \text{sgn}(x_0[i])$, and the subsequent inequality uses the fact that the product $x_r[i] \cdot \theta^*[i]$ is at most $x_r[i] \cdot (\theta_0[i] - \alpha \cdot \text{sgn}(x_0[i]))$ since the adversary's goal is to minimize this product and adversary's best response to $x_r[i]$ will do at least as well as the best response to $x_0[i]$ (which is a feasible, even if sub-optimal, response for the adversary).

We have shown that the inner product achieved by $x'$ would be greater than that of $x_r$, while the cost of $x'$ is also strictly less than $x_r$, since $x'$ keeps the $i$-th coordinate unchanged. Therefore, $\max_{\theta' \in \Theta_\alpha} J(x', \theta') < \max_{\theta' \in \Theta_\alpha} J(x_r, \theta')$, contradicting the assumption that $x_r \in \arg\min_{x' \in \mathcal{X}} \max_{\theta' \in \Theta_\alpha} J(x', \theta')$. A symmetric argument leads to a contradiction if we assume that $x_r[i] < x_0[i]$ even though $\theta_0[i] - \alpha \cdot \text{sgn}(x_0[i]) > 0$. $\square$

We now prove a lemma regarding the sequence of $|\theta'[i]|$ values of the dimensions that the while loop of Algorithms 1 changes.

**Lemma B.4.** *Let $j_k$ denote the dimension chosen in line 12 of Algorithm 1 during the $k$-th execution of its while-loop, and let $v_k$ denote the value of $|\theta'[j_k]|$ at a point in time (note that $\theta'$ changes over time). The sequence of $v_k$ values are decreasing with $k$.*

*Proof.* Note that in the $k$-th iteration of the while-loop, line 12 of Algorithm 1 chooses $j_k$ so that $j_k = \arg\max_{j \in \text{ACTIVE}} |\theta'[j]|$, based on the values of $\theta'$ at the beginning of that iteration. As a result, if $\theta'$ remains the same throughout the execution of the algorithm (which would happen if $\text{sgn}(x_r) = \text{sgn}(x_0)$, i.e., if none of the recourse coordinates changes from positive to negative or vice versa), then the lemma is clearly true. On the other hand, if the recourse "flips signs" for some dimension $i$, i.e., $\text{sgn}(x_r[i]) \neq \text{sgn}(x_0[i])$, this could lead to a change of the value of $\theta'[i]$. Specifically, as shown in Lemma B.2 and implemented in line 20 of the algorithm, the adversary changes $\theta'[i]$ to $\theta_0[i] + \alpha \cdot \text{sgn}(x_0[i])$. If that transition causes the sign of $\theta'[i]$ to change, then dimension $i$ becomes inactive and the algorithm will not consider it again in the future. If the sign of $\theta'[i]$ remains the same, then we can show that its absolute value would drop after this change, so even if it is considered in the future, it would still satisfy the claim of this lemma. To verify that its absolute value drops, assume that $x_0[i] > 0$, suggesting that the algorithm has so far lowered its value to 0, which would only happen if $\theta_0[i] < 0$ (otherwise, this change would be decreasing the inner product). Since $x_0[i] > 0$, the new value of $\theta'[i]$ is equal to $\theta_0[i] + \alpha$, and since this remains negative, like $\theta_0[i]$, we conclude that its absolute value decreased. A symmetric argument can be used for the case where $x_0[i] < 0$. $\square$

We are now ready to prove our main theoretical result (the proof of Theorem 3.3), showing that Algorithm 1 always returns an optimal robust recourse.

*Proof of Theorem 3.3.* To prove the optimality of the recourse $x_r$ returned by Algorithm 1, i.e., the fact that $x_r \in \arg\min_{x' \in \mathcal{X}} \max_{\theta' \in \Theta_\alpha} J(x', \theta')$, we assume that this is false, i.e., that there exists some other recourse $x^* \in \arg\min_{x' \in \mathcal{X}} \max_{\theta' \in \Theta_\alpha} J(x', \theta')$ such that $\max_{\theta' \in \Theta_\alpha} J(x^*, \theta') < \max_{\theta' \in \Theta_\alpha} J(x_r, \theta')$, and we prove that this leads to a contradiction.

Note that since $x^* \in \arg\min_{x' \in \mathcal{X}} \max_{\theta' \in \Theta_\alpha} J(x', \theta')$, it must satisfy Lemma B.3. Also, note that the way that Algorithm 1 generates $x_r$ also satisfies the conditions of Lemma B.3 (the choice of $\Delta$ in line 13 would never lead to a recourse of higher cost without improving the inner product), so we can conclude that if $x^*$ and $x_0$ were to change the same coordinate they would both do so in the same direction, i.e.,

$$\text{sgn}(x^*[i] - x_0[i]) = \text{sgn}(x_r[i] - x_0[i]).$$

Having established that for every coordinate $i$ the values of $x^*[i]$ and $x_r[i]$ will either both be at most $x_0[i]$ or both be at least $x_0[i]$, the rest of the proof performs a case analysis by comparing how far from $x_0[i]$ each one of them moves:

- **Case 1:** $\|x^* - x_0\|_1 = \|x_r - x_0\|_1$. Since $x^* \neq x_r$, it must be that $|x^*[i] - x_0[i]| > |x_r[i] - x_0[i]|$ for some $i$ and $|x^*[j] - x_0[j]| < |x_r[j] - x_0[j]|$ for some $j$. To get a contradiction for this case as well, we will consider an alternative recourse $x'$ that is identical to $x^*$ except for dimensions $i$ and $j$, each of which is moved $\delta$ closer to the values of $x_r[i]$ and $x_r[j]$, respectively, for some arbitrarily small constant $\delta > 0$. Formally,

$$x'[i] = x^*[i] + \delta \cdot \text{sgn}(x_r[i] - x^*[i]) \quad \text{and} \quad x'[j] = x^*[j] + \delta \cdot \text{sgn}(x_r[j] - x^*[j]).$$

Note that $x^*$ and $x'$ both have the same total cost since they only differ in $i$ and $j$ and

$$|x^*[i] - x_0[i]| + |x^*[j] - x_0[j]| = |x'[i] - x_0[i]| + \delta + |x'[j] - x_0[j]| - \delta$$
$$= |x'[i] - x_0[i]| + |x'[j] - x_0[j]|.$$

We let $\delta$ be small enough so that the adversary's response to $x^*$ and $x'$ is the same; for this to hold it is sufficient that a value of $x^*$ that is strictly positive does not become strictly negative in $x'$, or

vice versa. If we let $\theta'$ denote this adversary, then we have

$$x' \cdot \theta' - x^* \cdot \theta' = |(x'[j] - x^*[j]) \cdot \theta'[j]| - |(x'[i] - x^*[i]) \cdot \theta'[i]|$$
$$= \delta \cdot |\theta'[j]| - \delta \cdot |\theta'[i]|$$
$$= \delta \cdot (|\theta'[j]| - |\theta'[i]|),$$

where the first equality uses the fact that $x^*$ and $x'$ differ only on $i$ and $j$, and the fact that if we replace recourse $x^*$ with $x'$, then the change of $\delta$ on the $j$-th coordinate increases the distance from $x_0[j]$ and thus increases the inner product, while the change of $\delta$ on the $i$-th coordinate decreases the distance from $x_0[i]$ and thus decreases the inner product. The second equality uses the fact that the change on both coordinates $i$ and $j$ is equal to $\delta$.

To conclude with a contradiction, it suffices to show that $|\theta'[j]| > |\theta'[i]|$, as this would imply $x' \cdot \theta' > x^* \cdot \theta'$, contradicting the fact that $x^* \in \arg\min_{x' \in \mathcal{X}} \ \max_{\theta' \in \Theta_\alpha} J(x', \theta')$, since $x'$ would require the same cost as $x^*$ but it would yield a greater inner product. We consider three possible scenarios: *i)* If Algorithm 1 in line 12 chose to change dimension $i$ facing adversary $\theta'[i]$ before considering dimension $j$ and adversary $\theta'[j]$, then the fact that $|x^*[i] - x_0[i]| > |x_r[i] - x_0[i]|$ implies that the algorithm did not change coordinate $i$ as much as $x^*$ and it must have terminated after that via line 16; this would suggest that dimension $j$ and adversary $\theta[j]$ would never be reached after that, contradicting the fact that $|x^*[j] - x_0[j]| < |x_r[j] - x_0[j]|$. *ii)* If Algorithm 1 in line 12 chose to change dimension $j$ facing adversary $\theta'[j]$ and later on also considered dimension $i$ and adversary $\theta'[i]$, then Lemma B.4 suggests that $|\theta'[j]| > |\theta'[i]|$, once again leading to a contradiction. Finally, *iii)* if Algorithm 1 in line 12 chose to change dimension $j$ facing adversary $\theta'[j]$ and never ended up considering dimension $i$ even though $|\theta'[j]| < |\theta'[i]|$, this suggests that $i$ was removed from the ACTIVE set during the execution of the algorithm, which implies that $x_r[i] = 0$ and $|\theta_0[i]| < \alpha$, so moving further away from $x_0[i]$ would actually hurt the inner product because the adversary can flip the sign of $\theta'[i]$ via a change of $\alpha$. The fact that $x^*$ actually moved dimension $i$ further away then again contradicts the fact that $x^* \in \arg\min_{x' \in \mathcal{X}} \ \max_{\theta' \in \Theta_\alpha} J(x', \theta')$.

- **Case 2:** $\|x^* - x_0\|_1 < \|x_r - x_0\|_1$. In this case, we can infer that for some $i$ we have $|x^*[i] - x_0[i]| < |x_r[i] - x_0[i]|$, i.e., $x^*$ determined that the increase of the inner product achieved by moving $x^*[i]$ further away from $x_0[i]$ and closer to $x_r[i]$ was not worth the cost suffered by this increase. However, note that as we discussed in Observation B.1, $J(\cdot)$ is a decreasing function of the inner product. Also note that, since Algorithm 1 changes a coordinate of the recourse only if it increases the inner product, there must be some point in time during the execution of the algorithm when the inner product of $x' \cdot \theta'$ was at least as high as the inner product of $x^*$ with the adversarial response to $x^*$. Nevertheless, line 13 determined that this change would decrease the objective value $J(\cdot)$. If we specifically consider the last dimension $j$ changed by the algorithm, using Lemma B.4, we can infer that the value of $|\theta'[j]|$ at the time of this change was less than the value of $|\theta'[i]|$ for the dimension $i$ satisfying $|x^*[i] - x_0[i]| < |x_r[i] - x_0[i]|$; this is due to the fact that the algorithm chose to change $i$ weakly earlier than $j$. As a result, since line 13 determined that the increase of cost was outweighed by the increase in the inner product even though $|\theta'[j]| \leq |\theta'[i]|$, the inner product is greater, and $J(\cdot)$ is convex in the latter, this implies that increasing the value of $x^*[i]$ would also decrease the objective, thus leading to a contradiction of the fact that $x^* \in \arg\min_{x' \in \mathcal{X}} \ \max_{\theta' \in \Theta_\alpha} J(x', \theta')$.

- **Case 3:** $\|x^* - x_0\|_1 > \|x_r - x_0\|_1$. This case is similar to the one above, but rather than arguing that $x^*$ missed out on further changes that would have led to an additional decrease of the objective, we instead argue that $x^*$ went too far with the changes it made. Specifically, there must be some $i$ such that $|x^*[i] - x_0[i]| > |x_r[i] - x_0[i]|$, i.e., $x^*$ determined that the increase of the inner product achieved by moving $x^*[i]$ further away from $x_0[i]$ than $x_r[i]$ did was worth the cost suffered by this increase. Since the cost of $x^*$ is greater than the cost of $x_r$, it must be the case that its inner product is greater. Therefore, line 13 of the algorithm determined that moving $x_r[i]$ further away from $x_0[i]$ would not lead to an improvement of the objective even for a smaller inner product. Once again, the convexity of $J(\cdot)$ with respect to the inner product combined with the aforementioned facts implies that this increase must have hurt $x^*$ as well, leading to a contradiction. $\qquad\square$

## C  OMITTED DETAILS FROM SECTION 4

In this section, we provide additional results and analysis that were omitted from Section 4 due to space constraints. In Section C.1 we provide additional details on the running time of our algorithm

as well as what values were chosen for some of the hyper-parameters. Section C.2 adds more details on calculating the trade-off between robustness and consistency costs and provides error bars for Figure 1. Finally, Section C.3 details how parameter changes affect our results.

| Model | Dataset | $\lambda$ |
|-------|---------|-----------|
| LR | Synthetic Data | 1.0 |
| | German Credit Data | 0.5 - 0.7 |
| | Small Business Data | 1.0 |
| NN | Synthetic Data | 1.0 |
| | German Credit Data | 0.1 - 0.2 |
| | Small Business Data | 1.0 |

Table 1: $\lambda$ that maximize the validity with respect to the original model $\theta_0$ for each dataset. The other choices of parameters are mentioned in Section C.1.

## C.1 ADDITIONAL EXPERIMENTAL DETAILS

The experiments were conducted on two laptops: an Apple M1 Pro and a 2.2 GHz 6-Core Intel Core i7. Our algorithm that generates results for the robustness versus consistency trade-off takes 45-60 minutes to run to generate each of the subfigures in Figure 1.

In our robustness versus consistency experiments in Section 4.1, we choose $\alpha = 0.5$ and find the $\lambda$ that maximizes the validity with respect to the original model $\theta_0$ in each round of cross-validation. The range of $\lambda$ values found to maximize the $\theta_0$ validity for each setting is reported in Table 1.

In our experiment on smoothness in Section 4.1, we created the future model using a modified dataset similar to Upadhyay et al. (2021). To produce the altered synthetic data, we employed the same method outlined in Section 4, but we changed the mean of the Gaussian distribution for class 0. The new distribution is $x \sim N(\mu'_0, \Sigma_y)$, where $\mu'_0$ is equal to $\mu_0 + [\alpha, 0]^T$, while $\mu'_1$ remained unchanged at $\mu_1$. We used this new distribution to learn a model for the correct prediction. The German credit dataset Hofmann (1994) is available in two versions, with the second one Grömping (2020) fixing coding errors found in the first. This dataset exemplifies a shift due to data correction. We used the second dataset to learn the model for the correct prediction. The Small Business Administration dataset Min Li & Taylor (2018), which contains data on 2,102 small business loans approved in California from 1989 to 2012, demonstrates temporal shifts. We split this dataset into two parts: data points before 2006 form the original dataset, while those from 2006 onwards constitute the shifted dataset. We used the shifted dataset to learn a model for the correct prediction.

To generate the predictions in our smoothness experiment in Section 4.1, we define $\epsilon$ as half the distance between the original model $\theta_0$, and the shifted model $\hat{\theta}_*$ which we use as the correct prediction for the future model. Perturbations of $\pm\epsilon$ and $\pm 2\epsilon$ are then applied to each dimension of the $\hat{\theta}_*$. For linear models, we use $\epsilon = 0.12$ for the Synthetic dataset, $\epsilon = 0.16$ for the German dataset, and $\epsilon = 0.43$ for the Small Business Administration dataset. For non-linear models, the amount of perturbation is determined by each instance in the dataset by using the LIME approximation to provide recourse. More details can be found in our code. In all cases, the perturbed values are clamped to ensure they remain within the $\alpha = 1$ in terms of $L^1$ distance from the original model $\theta_0$.

In our cost versus worst-case validity experiments in Section 4.2, we set $\alpha = 0.2$ and employ projected gradient ascent to identify a single worst-case predictive model. During each iteration of projected gradient ascent, the model's weights and biases are constrained within the range of the initial model $\theta_0$ plus or minus $\alpha$. The optimization is performed using the Adam optimizer with a learning rate of 0.001, and the binary cross entropy loss function is utilized.

## C.2 ROBUSTNESS-CONSISTENCY TRADE-OFF

In this section, we first provide more details on how we solve for $\arg\min_{x'} \max_{\theta' \in \Theta_\alpha} \beta \cdot J(x', \theta') + (1 - \beta) \cdot J(x', \hat{\theta})$ for varying $\beta \in [0, 1]$. We use an approach similar to Algorithm 1, with modifications to the selection process for the next coordinate $i$ to update. Our objective is to identify the recourse that satisfies $\arg\min_{x'} \max_{\theta' \in \Theta_\alpha} \beta \cdot J(x', \theta') + (1 - \beta) \cdot J(x', \hat{\theta})$. In each iteration of the while loop, instead of choosing the coordinate with the maximum absolute value of $\theta'$, we determine $i$ by solving $i \in \arg\min_j \beta \cdot J(x' + \Delta e_j, \theta') + (1 - \beta) \cdot J(x' + \Delta e_j, \hat{\theta})$ where $\Delta e_j$ is identified using grid search.

We then provide figures that are identical to Figure 1 but also contain error bars. Figures 4 and 5 replicates Figure 1 but also include error bars. These error bars are calculated for the robustness (Figure 4) and consistency costs (Figure 5) when averaging is done over all the data points in the test set that require recourse as well as the folds.

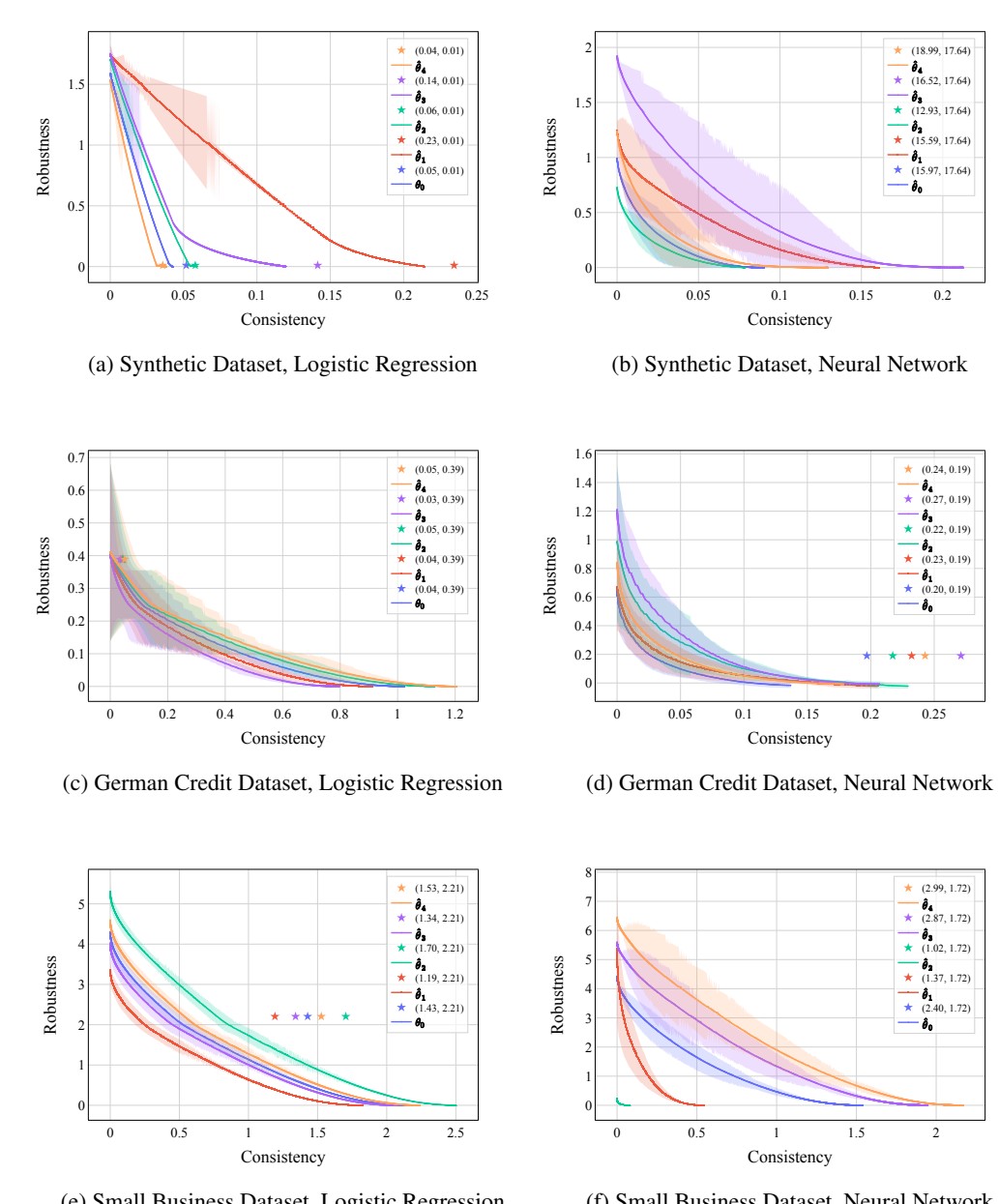

(a) Synthetic Dataset, Logistic Regression

(b) Synthetic Dataset, Neural Network

(c) German Credit Dataset, Logistic Regression

(d) German Credit Dataset, Neural Network

(e) Small Business Dataset, Logistic Regression

(f) Small Business Dataset, Neural Network

Figure 4: The Pareto frontier of the trade-off between robustness and consistency for $\alpha = 0.5$ with error bars for robustness: logistic regression (left) and neural network (right). Rows correspond to datasets: synthetic (top), German (middle), and Small Business (bottom). In each subfigure, each curve shows the trade-off for different predictions as mentioned in the legend. The robustness and consistency of the recourse solution found by ROAR are mentioned in the parentheses and depicted by stars. Missing stars are outside of the range of the coordinates of the figure.

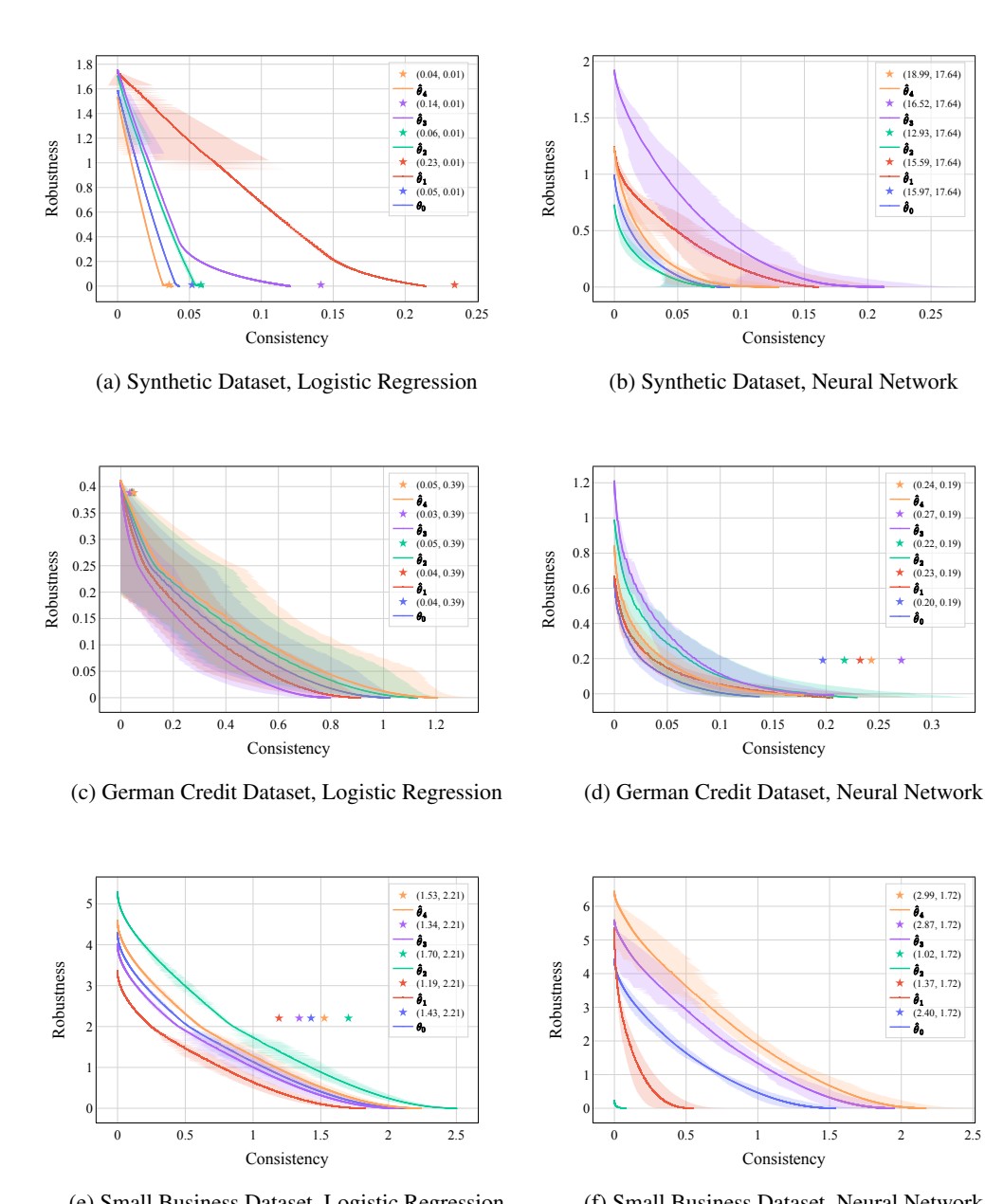

Figure 5: The Pareto frontier of the trade-off between robustness and consistency for $\alpha = 0.5$ with error bars for consistency: logistic regression (left) and neural network (right). Rows correspond to datasets: synthetic (top), German (middle), and Small Business (bottom). In each subfigure, each curve shows the trade-off for different predictions as mentioned in the legend. The robustness and consistency of the recourse solution found by ROAR are mentioned in the parentheses and depicted by stars. Missing stars are outside of the range of the coordinates of the figure.

## C.3 EFFECT OF THE PARAMETERS

In the experiments on the trade-off between robustness and consistency in Section 4.1 we used $\alpha = 0.5$ and a $\lambda$ that maximizes the validity of recourse with respect to this $\alpha$. In this section, we see how varying $\alpha$ can affect the results. In particular, in Figures 6 and 7, we replicated the trade-offs presented in Figure 1 in Section 4.1 with $\alpha = 0.1$ and $\alpha = 1$, respectively. Again, for each choice of $\alpha$, we selected a $\lambda$ that maximizes the validity

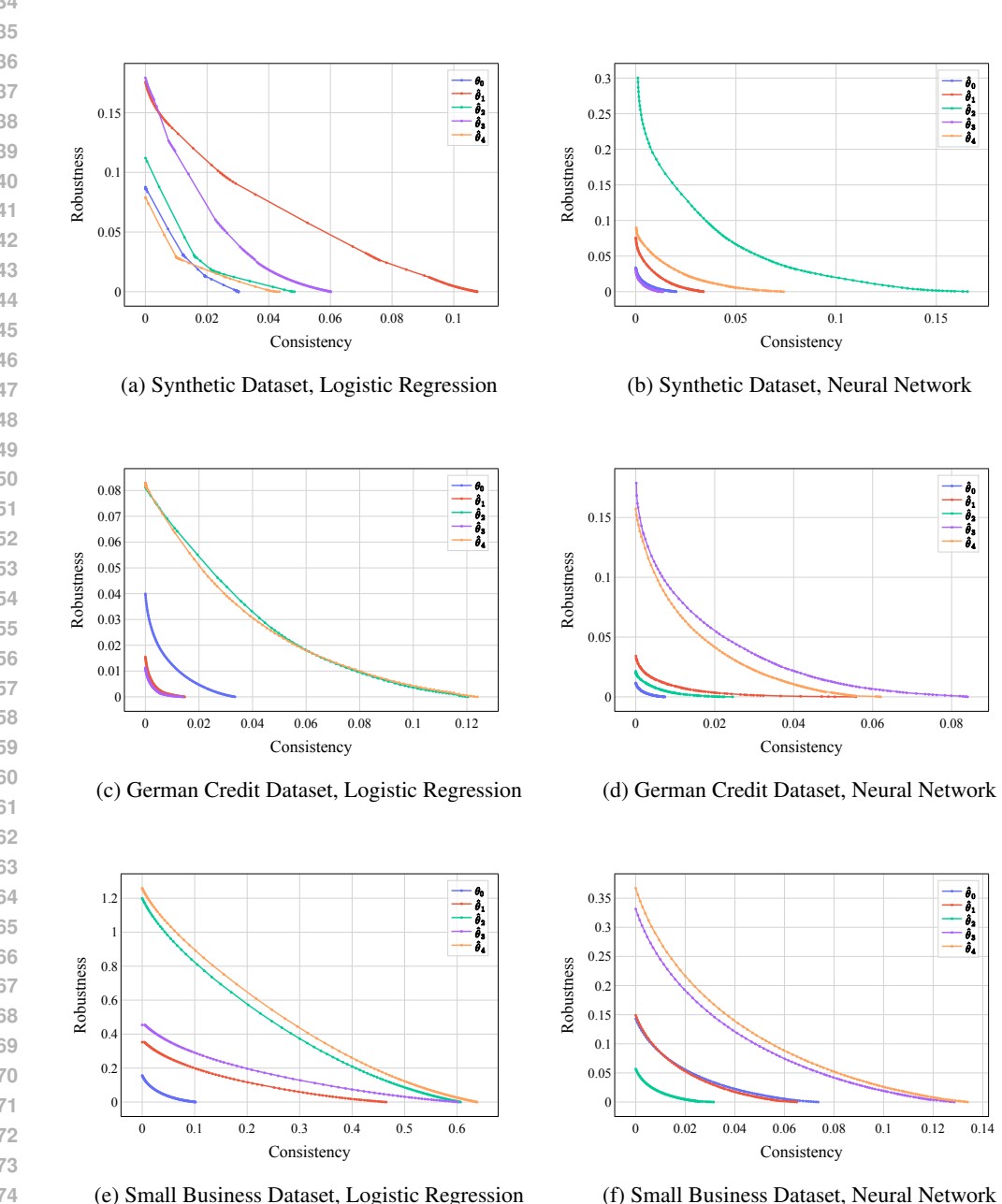

Figure 6: The Pareto frontier of the trade-off between robustness and consistency for $\alpha = 0.1$: logistic regression (left) and neural network (right). Rows correspond to datasets: synthetic (top), German (middle), and Small Business (bottom). In each subfigure, each curve shows the trade-off for different predictions as mentioned in the legend.

of recourse with respect to this $\alpha$. We generally observe that increasing $\alpha$ increases both the robustness and consistency costs.

## C.4 EXPERIMENTS ON LARGER DATASETS

In this section, we provide experimental results for a dataset that is much larger (both in terms of number of instances and number of features) compared to the datasets in Section 4. The running time of our algorithm scales linearly with the number of instances for which recourse is provided. For each instance, the running time

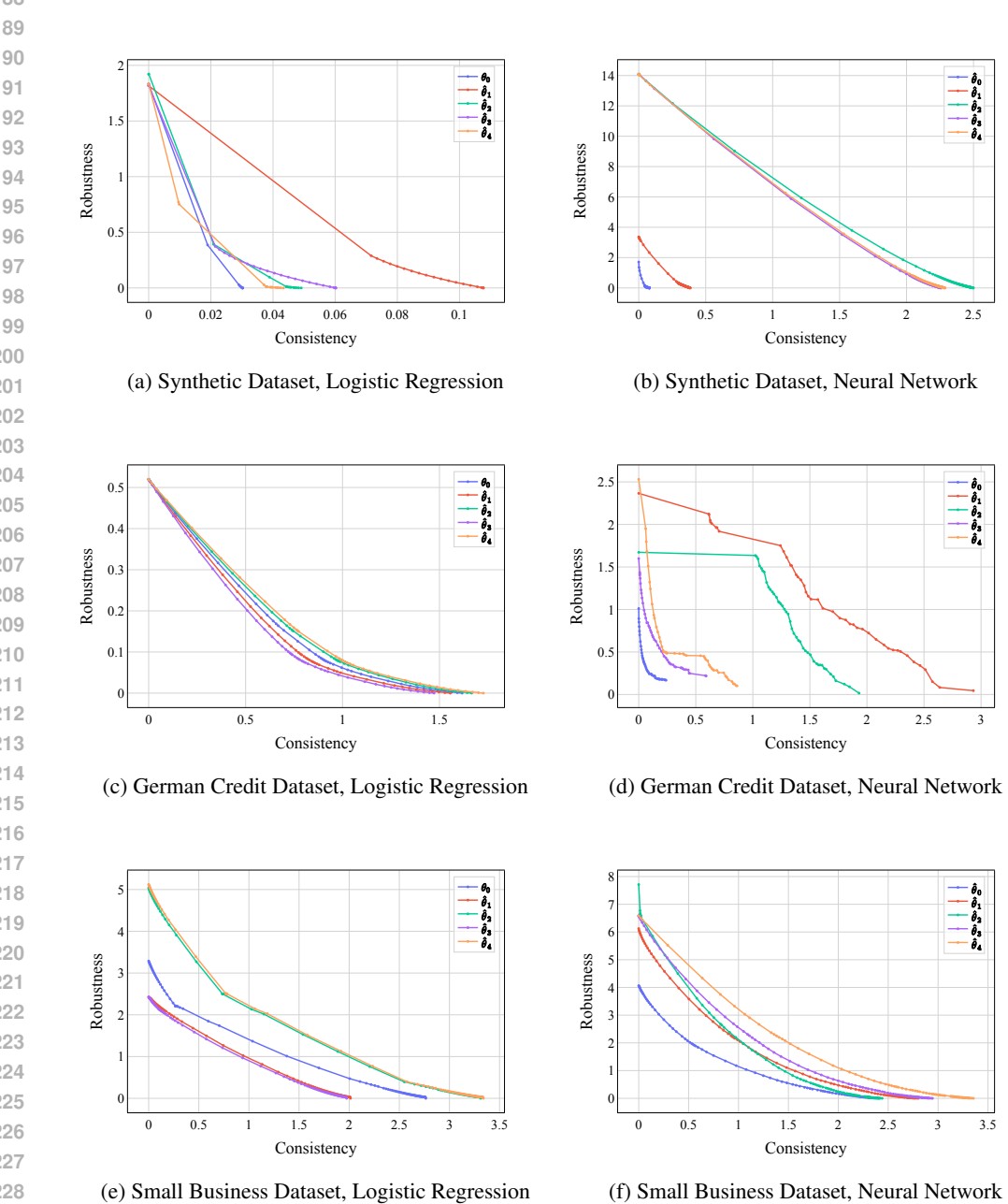

(a) Synthetic Dataset, Logistic Regression

(b) Synthetic Dataset, Neural Network

(c) German Credit Dataset, Logistic Regression

(d) German Credit Dataset, Neural Network

(e) Small Business Dataset, Logistic Regression

(f) Small Business Dataset, Neural Network

Figure 7: The Pareto frontier of the trade-off between robustness and consistency for $\alpha = 1$: logistic regression (left) and neural network (right). Rows correspond to datasets: synthetic (top), German (middle), and Small Business (bottom). In each subfigure, each curve shows the trade-off for different predictions as mentioned in the legend.

of our algorithm grows linearly in the number of features since the minimization problem in Line 13 of our algorithm can be solved analytically. For non-linear models, the cost of approximating the model with a linear function should be added to the total cost per instance.

We use the ACSIncome-CA (Ding et al., 2021) dataset for experiments in this section. This dataset originally consisted of 195,665 data points and 10 features, 7 of which are categorical and have been one-hot encoded. However, to lower the runtime, we sub-sampled the dataset to include 50,000 data points, and removed the categorical feature "occupation (OCCP)", as it contains more than 500 different occupations. This left us with more than 250 features after one hot encoding.

Figure 8 depicts the trade-off between the cost and validity of recourse for both logistic regression and neural network models. The choices of parameters used for results in Figure 8 are the same as the results for Figure 3 in Section 4.2. We observe that even in a dataset with a much larger number of features, Algorithm 1 can generate recourses with high validity, especially for logistic regression models. Similar to Figure 3, achieving very high validity comes at a cost of higher implementation cost which is higher than the cost required for smaller datasets. See Figure 3.

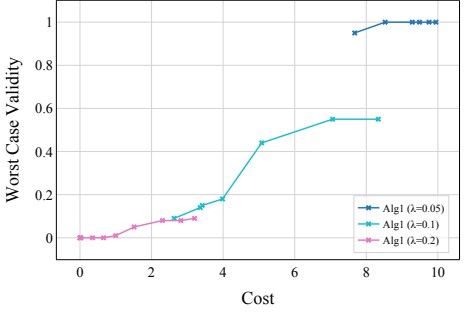
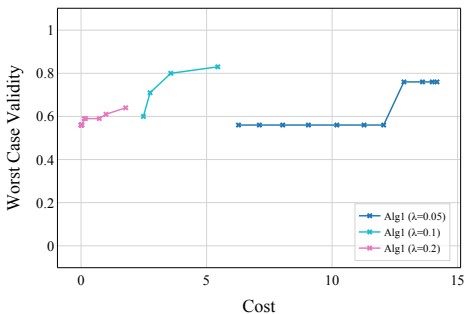

(a) ACS Income Dataset, Logistic Regression          (b) ACS Income Dataset, Neural Network

Figure 8: The Pareto frontier of the trade-off between the worst-case validity and the cost of recourse solutions. The left panel is for the logistic regression while the right panel is for a 3-layer neural network for the ACS Income dataset. In each subfigure, each curve shows the trade-off for different methods as mentioned in the legend.

