# OpenReview forum: "Learning-Augmented Robust Algorithmic Recourse"
_ICLR.cc/2025/Conference — Submitted to ICLR 2025_

### Official Review · Reviewer_dbwt · 2024-10-30

**Soundness:** 2
**Presentation:** 3
**Contribution:** 2
**Rating:** 3
**Confidence:** 4

**Summary:**

The paper studies the problem of robustness of algorithmic recourse in the context of model shifts. The authors propose an algorithm to optimize a well-established adversarial min-max objective in the context of generalized linear models. Additionally, they consider the setting in which we have access to an estimate of the model change $\hat{\theta}$ and show there is a trade-off between robustness and consistency in several experiments. They also perform some validation considering two baselines (ROAR and RBR).

**Strengths:**

The paper tackles a relevant problem of the recourse literature, namely, the robustness of counterfactual explanations to model shifts. The paper is relatively easy to follow, and it focuses on studying a simple yet potentially important phenomenon arising from the trade-off between consistency and robustness (or regret).

**Weaknesses:**

I have some concerns about the novelty and experimental analysis of the proposed approach. In general, some design choices (e.g., assuming to know the model shift), are interesting, but they could be better explored.

**[Contributions]**  In Section 1, the statement “the first optimal algorithm for any robust recourse problem” is misleading. There are other settings in which recourse robustness is required in which your method might **not** achieve the optimal (see [1] for an exhaustive list). Moreover, one could argue that in practice most models are deep neural networks, thus making the paper’s approach lose any guarantees.

**[Learning-Augmented Framework]** The authors consider a setting in which they have access to a prediction over the new model parameters $\hat{\theta}$. In practice, one could simply compute solutions for robust recourse using standard methods (e.g., ROAR, RBR) by considering the updated model $f_\hat{\theta}$. This consideration undermines the overall contribution to the approach (since both objective (3) and metrics (5), (6) are very similar to related cited literature). I recommend that the authors discuss the limitations/issues arising from obtaining a prediction of the model update and/or propose a computational model to describe them better.

**[Algorithm 1]** Section 3.1 would benefit a computational complexity analysis of Algorithm 1. It might as well achieve the global optimum, but if it is exponential in the number of features we could settle for approximate solutions. Lastly, in the case of a local approximation, Algorithm 1 might not be optimal anymore, and the authors should discuss the limitations of their approach in such a scenario.

**[Experimental Analysis with $\hat{\theta}$]**
- Following the [**Learning-Augmented Framework**] comment, it is not clear if the analysis in Section 4 considers competitors (ROAR and RBR) using the predicted model $f_\hat{\theta}$ or the original $f_\theta$. The latter case would penalize the competitors and it would result in an unfair comparison between the approaches.
- In Section 4 (Figure 2), the authors consider a simple setting in which the $\alpha$ parameter in Algorithm 1 is the same used to perturb the model. I believe it is a simplifying assumption since in practice we might have access to a noisy estimator of $\alpha$.
- The experiments could benefit from additional baselines using adversarial min-max objectives (e.g., Dominguez-Olmedo et al., 2022 can be run without the causal structure, by only considering the $\epsilon$-ball) or other forms of robustness (see Section 4 of [1] for a fairly large list).

[1] Jiang, Junqi, et al. "Robust counterfactual explanations in machine learning: A survey." IJCAI (2024).

**Questions:**

- Could you provide a computational complexity analysis for Algorithm 1?
- What if the $\alpha$ you consider as perturbation in the experiments (Figure 2) is different from the actual $\alpha$ used by Algorithm 1?
- Do you compute recourses with ROAR and RBR by considering the model with parameters $\hat{\theta}$ or the original $\theta_0$?
- How do you evaluate the recourses computed on the LIME surrogate? Do you consider the validity of the original non-linear model $f_\theta$?

---

> ### Author Response · Authors · 2024-11-22
> **Response to Reviewer dbwt**
>
> We thank the reviewer for their comments and questions. Below we provide answers to the reviewer’s questions and concerns.
>
>
> **First optimal algorithm for robust recourse:** Thank you for pointing out the survey on robust recourse. We will cite this survey in the next version. The problem of computing robust recourse (even for linear models and our given modeling choice) is non-convex. We reviewed the material in the survey and still could not find any work that provides a *computationally efficient* algorithm for computing *optimal* robust recourse. Given the high confidence of the reviewer, we would appreciate it if the reviewer could point us to any paper that provides such an algorithm.
>
>
> **Computational complexity of Algorithm 1:** The running time of Algorithm 1 is mentioned in line 1187 which we restate here: “The running time of our algorithm scales linearly with the number of instances for which recourse is provided. For each instance, the running time of our algorithm grows linearly in the number of features since the minimization problem in Line 13 of our algorithm can be solved analytically. For non-linear models, the cost of approximating the model with a linear function should be added to the total cost per instance.” We would be happy to emphasize this more clearly in Section 3.
>
>
> **Algorithm guarantees and extension to other settings** Although the algorithm that we provide is indeed the first efficient algorithm for computing the optimal robust recourse for the linear model (a non-convex problem!), our results do evaluate this algorithm in other settings, like the ones suggested by the reviewer. For example, the experiments in Section 4.2 highlight the power of our algorithm even for nonlinear (neural network/MLP) functions by first approximating these functions using a linear function and then applying our optimal algorithm to the approximation. As we show, this provides comparable and sometimes better cost-validity trade-offs compared to the state-of-the-art robust recourse approaches. However, we would like to emphasize again that the main focus of our paper is on the use of learning-augmented solutions in this setting; the fact that we also provide an optimal algorithm even for purely adversarial settings is the additional contribution that allows us to provide an even more emphatic improvement over the prior work on robust algorithmic recourse.
>
>
> **Comparison with ROAR and RBR:** Neither ROAR nor RBR utilize the prediction. So the only information available about the future model is the original model $\theta_0$ and how much this model can change ($\alpha$). Note that our comparisons with both ROAR and RBR in Section 4.2 are only based on the breakdown of the total cost of computing robust recourse. Such comparisons do not involve any predictions (i.e., all the algorithms solve the same optimization problem) so our algorithm does not have any unfair advantages compared to ROAR and RBR in experiments in Section 4.2.
>
>
> **Validity on LIME surrogates:** For non-linear models, we use our algorithm to compute the recourse on the linear approximation of the model which is generated by LIME. Even though the recourse is computed using this approximation, we still measure the validity with respect to the non-linear model, not the approximation. This way of measuring validity allows us to fully demonstrate the power of our approach in achieving better cost-validity trade-offs compared to the state-of-the-art robust recourse approaches.
>
>
> **Additional baselines for the experiments** Other baselines for robust recourse either assume a specific model class (such as decision trees) or alternate ways of model change. We see ROAR and RBR as the only two valid baselines as the robust recourse formulation for them and ours solves the same problem (same model class, cost function, and model change).

---

> > ### Comment · Reviewer_dbwt · 2024-11-25
> >
> > I would like to thank the authors for their reply.
> >
> > **[Comparison with ROAR and RBR and Learning-Augmented Framework]** Given that neither ROAR nor RBR utilizes the prediction $\hat{\theta}$ and that neither Algorithm 1 does as far as I understood from the pseudocode, the significance of the learning-augmented framework for recourse is still not clear to me.
> >
> > However, the authors mention that they use a variant of Algorithm 1 (line 331) to solve a min-max problem using the prediction $\hat{\theta}$. If that is the case, the most natural comparison is using ROAR with $\hat{\theta}$, rather than $\theta_0$ since it is the best we can do (Section 4.1). I also do not understand the rebuttal phrase "[...] our comparisons with both ROAR and RBR in Section 4.2 are only based on the breakdown of the total cost [...]" since in lines 496-497 they mention the trade-off between the worst-case validity and cost of recourse.
> >
> > As far as I understood, the authors propose two algorithms: Algorithm 1 (to solve Equation 3) and another Algorithm 1 that is used to solve the learning-augmented framework (Appendix C). However, these algorithms can be two orthogonal contributions, and I believe the paper does not help in understanding clearly the contributions or the experimental evaluations. By looking at the other comments, I believe this confusion is shared by the other reviewers.
> >
> > I would suggest the authors focus on one of these contributions and propose it as the main one to improve the clarity of their message. I will keep my score since I believe the paper needs some additional iterations before being accepted.

---

> > > ### Author Response · Authors · 2024-11-25
> > >
> > > We appreciate the reviewer’s engagement during the discussion period and for pointing out what the source of the confusion may be. Let us try to explain this, since we believe that minor edits and restructuring in the writing of our submission could alleviate this issue without affecting the results.
> > >
> > >
> > > Our contribution is, indeed, two-fold:
> > >
> > > (1) We propose and analyze Algorithm 1, which does not use any predictions as input. This algorithm improves upon the algorithms used in prior work on robust algorithmic recourse. Prior work used algorithms that could get stuck in local optima even for linear models, while our algorithm avoids this issue and quickly finds global optima. As a result, this is a contribution that is valuable even beyond the learning-augmented model.
> > >
> > > (2) Moving beyond the previously studied robust algorithmic recourse problem, we then adapt the learning-augmented model to evaluate the extent to which recourse algorithms that are augmented with unreliable predictions regarding the future can simultaneously achieve good robustness (i.e., good performance if the model is generated adversarially, just like in the prior robust algorithmic recourse work) and good consistency (i.e., good performance if the prediction happens to be accurate). This provides a new framework for designing and evaluating algorithms using two criteria rather than one. To evaluate the extent to which this is possible, we consider a family of algorithms that take as input an unreliable prediction regarding the future model as well as a parameter $\beta$ and then use a variation of Algorithm 1 to optimize the objective of Lines 330-331 (the algorithm itself appears in Appendix C as the reviewer pointed out). The plots of Figure 1 exhibit how these algorithms perform with respect to these two criteria (robustness and consistency) in a variety of different settings. Figure 2 goes even further and evaluates the performance of these algorithms not only for adversarial predictions or accurate predictions but, instead, as a function of the prediction error.
> > >
> > >
> > > It is important to note that the plots of Figure 1 not only provide us with an evaluation of our algorithms in the newly introduced learning augmented setting, but they also provide a very direct comparison to prior work on the standard robust algorithmic recourse problem. Specifically, the only goal of this prior work was to minimize the robustness, without worrying about consistency. In Figure 1, we include the robustness from ROAR using the y-coordinate of the stars (you can disregard their x-coordinate). One of the interesting results that Figure 1 exhibits quite emphatically is that our algorithms do not just beat the robustness of prior work (we can achieve a y-coordinate of 0 by setting the parameter $\beta=1$); they even combine better robustness than this prior work with strong consistency guarantees: e.g., in the dataset of Figure 1(b) note that the robustness of ROAR is equal to 17.64, while our algorithms can achieve a perfect consistency of zero for a variety of different predictions with a robustness that is no more than 2! We can therefore simultaneously beat the best-known robustness from prior work while also guaranteeing optimal performance in case the unreliable prediction happens to be accurate. We consider this to be a compelling observation, and the other figures exhibit analogous results.
> > >
> > >
> > > Due to space limitations, we had to really compress our discussion regarding the experimental results, but we would be happy to generate a revision of the paper that more accurately captures the description above. We believe the cause of the confusion is not the set of results, but the structure of the presentation.

---

> > > > ### Comment · Reviewer_dbwt · 2024-11-29
> > > >
> > > > I thank the authors for their answers who cleared some of my doubts. However, I am still of the opinion that there are serious issues with the presentation and the relevance of the "learning-augmented framework". For example:
> > > >
> > > > - In the "Abstract" and "Introduction", the authors focus on the _learning-augmented framework_, while in the next sections, they provide a solution for the robust algorithmic recourse problem which is a _different and orthogonal_ problem.
> > > > - In the "Experiments" section, there is no real emphasis on the "model changes", or how they relate to the concept of "prediction" of the _learning-augmented framework_.
> > > > - In Section 4.1, the experiments look at the robustness/consistency trade-off, but the _learning-augmented framework_ seems to have little part in the evaluation (e.g., the "Smoothness" paragraph considers only permutations over $\theta$, so it is even more unclear what the framework is for).
> > > > - In Section 4.1, the algorithm considered is the one in the Appendix, not the one presented in the main text, thus adding more confusion.
> > > >
> > > > Since the information is scattered around the paper, it becomes difficult to understand the main contribution and the relevance of the _learning-augmented framework_. For example, as I mentioned in my initial review, if I have a prediction $\hat{\theta}$, I could think about using ROAR on this updated model $f_\hat{\theta}$, and if $f$ is non-linear and the prediction is accurate, ROAR might perform much better than the proposed approach which works only for linear models. Such a possibility is not considered in the experiments.
> > > >
> > > > Unfortunately, I do not think these issues can be resolved with minor edits, but it would require a serious reworking of the manuscript. To be completely honest, it also becomes harder for me as a reviewer to suggest actionable improvements. In practice, I would suggest the authors review the manuscript and focus more on either (a) the optimal robust solution, or (b) how to integrate the "learning-augmented framework" within the recourse. In turn, it will leave more space to present the empirical evaluation more thoroughly which currently is also an issue (as acknowledged by the authors).

---

### Official Review · Reviewer_fFjE · 2024-11-03

**Soundness:** 2
**Presentation:** 1
**Contribution:** 2
**Rating:** 5
**Confidence:** 4

**Summary:**

In this paper, the author proposes a learning-augmented method for generating algorithmic recourse by accounting for the dynamic behavior of the target model. They assume that the designer has access to an unreliable prediction model, which can be leveraged to improve upon prior robust recourse methods. By identifying an anchor using the unreliable model, they assess the consistency between the generated recourse and this anchor. The problem then shifts to finding an optimal balance between robustness and consistency.

**Strengths:**

1 The topic is quite interesting. It points out an interesting topic of algorithmic recourse.
2 The algorithm is concise and a theoretical guarantee is provided.
3 Several experiments are implemented.

**Weaknesses:**

1 The presentation lacks clarity. Although the innovation is highlighted at the beginning, it’s unclear why the learning-augmented framework is proposed as the solution to this issue. There seems to be a logical gap in Section 3 that needs to be addressed.
2 Their method is quited limited, as it only considers the case of linear model.
3 In Section 3, they mention that the prediction model is unreliable, yet it is still used to measure the consistency of the algorithmic recourse. However, they do not explain how the uncertainty of the prediction model is measured or the extent to which this uncertainty impacts the solution.

**Questions:**

1  Could you further strenthen the intuition of your method? I suggest that the author provide a clear reason for adopting the learning-augmented framework for this task and include more information in the related work section.
2 Coud you add a case study to demonstrate the advantages of the proposed method?
3 If possible, could you consider a more general case of the target model, e.g. MLP?

---

> ### Author Response · Authors · 2024-11-22
> **Response to Reviewer fFjE**
>
> We thank the reviewer for their comments and questions. However, all three weaknesses that the reviewer listed suggest that they did not read our submission very closely and have some fundamental misunderstandings regarding even some of the central notions. We ask that the reviewer take the time to more carefully read our submission along with this rebuttal in order to properly evaluate our results.
>
>
> **Logical gap in Section 3:** We would greatly appreciate it if the reviewer could elaborate on what this “logical gap” that they are referring to may be. Section 3 formulates the problem of learning-augmented robust recourse, it formally defines the notions of consistency and robustness, which are adaptations of standard performance benchmarks in the learning-augmented framework, and then provides an algorithm to compute robust and consistent recourse. The results of this section are then used in our subsequent experiments. If the reviewer is unsure about the contribution of this section, we are concerned that they may have misunderstood the contribution of our submission, and we would happily clarify further.
>
>
> **Method is limited and extension to other settings such as MLP:** Our results are actually **not** restricted to the linear case. Although the algorithm that we provide is indeed the first efficient algorithm for computing the optimal robust recourse for the linear model (a non-convex problem!), our results do evaluate this algorithm in other settings, like the ones suggested by the reviewer. For example, the experiments in Section 4.2 highlight the power of our algorithm even for nonlinear (neural network/MLP) functions by first approximating these functions using a linear function and then applying our optimal algorithm to the approximation. As we show, this provides comparable and sometimes better cost-validity trade-offs compared to the state-of-the-art robust recourse approaches. However, we would like to emphasize again that the main focus of our paper is on the use of learning-augmented solutions in this setting; the fact that we also provide an optimal algorithm even for purely adversarial settings is the additional contribution that allows us to provide an even more emphatic improvement over the prior work on robust algorithmic recourse.
>
>
> **Unreliable predictions and consistency:** Unfortunately, we think the reviewer does not have an accurate understanding of the notion of “consistency,” which is a technical term that we formally introduce in Section 3. Consistency is a standard notion in the learning-augmented framework, used in order to measure the performance of the algorithm in the case that the prediction is correct. Whenever the prediction is inaccurate, the performance is instead evaluated using the “robustness” notion. The goal in this literature is to simultaneously achieve good robustness and consistency. So, while the framework allows the prediction to have errors, consistency is still a very well-defined measure. We also provide results regarding smoothness, which interpolates between these two notions.
>
>
> **Could you add a case study to demonstrate the advantages of the proposed method?** We are, once again, quite unsure about what the reviewer means with this comment. Our experimental section is dedicated to exhibiting the advantages of our proposed method over a variety of datasets.
>
>
> **Why is the learning-augmented framework proposed?** We are honestly puzzled by the reviewer’s statement that “Although the innovation is highlighted at the beginning, it’s unclear why the learning-augmented framework is proposed as the solution to this issue.” The main goal of our paper is to use (potentially unreliable) predictions to alleviate the high cost that comes with previously introduced solutions for robust algorithmic recourse. We dedicated all of Section 4.1 to evaluate the benefits of this approach by measuring not only the trade-off between robustness and consistency but also the smoothness (the performance as a function of the prediction error). These results exhibit how we can utilize even unreliable predictions to significantly reduce the total cost of robust recourse. This would have clearly been impossible without the adaptation of the learning-augmented framework and the robustness and consistency notions. We are therefore unable to see what the reviewer means to say with this statement and our best guess is that it may be due to a misunderstanding which this response hopefully addressed. If not, we would be happy to provide more information regarding the central role that this framework plays in our paper.

---

### Official Review · Reviewer_ERQu · 2024-11-07

**Soundness:** 3
**Presentation:** 2
**Contribution:** 1
**Rating:** 5
**Confidence:** 4

**Summary:**

The authors study in this paper the notion of robust recourse. A recourse is defined in the first paragraph: it ``provides each individual who was given an undesirable label with a minimum cost improvement suggestion to achieve the desired label”. The problem studied here is that the classifier may change (for example, it could be updated as new data arrive), which can invalidate the recourse.

The paper subsequently consider a recourse found by solving a min-max problem: the minimization is over the space of input feature, the maximization is over the perturbation set of the model parameter. The authors propose Algorithm 1, which is proved to converge to the global optimal solution under specific condition. The results are supported empirically by the numerical experiments with three real world datasets.

**Strengths:**

1. The paper proposed Algorithm 1, which can compute the robust recourse to global optimality, despite the non-convexity of the robust recourse problem.

**Weaknesses:**

1. The guarantee is only for a generalized linear model, a 1-norm cost on $x$ and an $\infty$-norm neighborhood for $\theta$.

2. It is unclear how the insights from the learning-augmented framework has impacted the main contributions of this paper. I feel that I can omit all the sentences discussing learning-augmented framework without changing meaning/results/implications of the paper.

3. Despite the positive results for the case presented in the paper, it is unclear how the result can be extended to any other settings (different cost, or nonlinear model without linear surrogate, etc.)

**Questions:**

1. Is there any convergence guarantee for the algorithm in Section C2? If there is no convergence guarantee, then how could we interpret the results in the numerical experiments? Could we attribute the superior performance of the recourse to the algorithm or to the different loss function?

2. Please provide out-of-sample validity results. In the experiment settings of ROAR, the out-of-sample can simulate different shifts of the ML models. The out-of-sample validity computed this way is a more practical criterion to compare different methods. In fact, the set $\Theta_\alpha$ is only a proxy to model our belief about future model perturbation, and it has no real meaning when we compute the worst-case validity.

---

> ### Author Response · Authors · 2024-11-22
> **Response to Reviewer ERQu**
>
> We thank the reviewer for their comments and questions. Below we provide answers to the reviewer’s questions and concerns.
>
>
> **Insights from learning-augmented framework:** We are honestly puzzled by the reviewer’s statement that “all the sentences discussing learning-augmented framework without changing meaning/results/implications of the paper.” The main goal of our paper is to use (potentially unreliable) predictions to alleviate the high cost that comes with previously introduced solutions for robust algorithmic recourse. We dedicated all of Section 4.1 to evaluate the benefits of this approach by measuring not only the trade-off between robustness and consistency but also the smoothness (the performance as a function of the prediction error). These results exhibit how we can utilize even unreliable predictions to significantly reduce the total cost of robust recourse. This would have clearly been impossible without the adaptation of the learning-augmented framework and the robustness and consistency notions. We are therefore unable to see what the reviewer means to say with this statement and our best guess is that it may be due to a misunderstanding which this response hopefully addressed. If not, we would be happy to provide more information regarding the central role that this framework plays in our paper.
>
>
> **Convergence guarantee for the algorithm in Section C2:** We can thankfully directly address the reviewer’s concern regarding the convergence of this algorithm. This is a greedy algorithm similar to Algorithm 1, with slight modifications on how to pick the coordinate to change and how much to change the chosen coordinate. These changes do not change the asymptotic running time or convergence guarantee of this algorithm compared to Algorithm 1, so the algorithm in Section C2 runs in time linear in the number of dimensions. See line 1187 for a more detailed running time analysis of Algorithm 1. We would be happy to make this more explicit in the next revision of our paper.
>
>
> **How the result can be extended to other settings:** Although the algorithm that we provide is indeed the first efficient algorithm for computing the optimal robust recourse for the linear model (a non-convex problem!), our results do evaluate this algorithm in other settings, like the ones suggested by the reviewer. For example, the experiments in Section 4.2 highlight the power of our algorithm even for nonlinear (neural network) functions by first approximating these functions using a linear function and then applying our optimal algorithm to the approximation. As we show, this provides comparable and sometimes better cost-validity trade-offs compared to the state-of-the-art robust recourse approaches. However, we would like to emphasize again that the main focus of our paper is on the use of learning-augmented solutions in this setting; the fact that we also provide an optimal algorithm even for purely adversarial settings is the additional contribution that allows us to provide an even more emphatic improvement over the prior work on robust algorithmic recourse.

---

### Meta-Review · Area_Chair_Yz28 · 2024-12-20

**Metareview:**

This paper aims to address the robust recourse problem, where a model might update over time invalidating the recourse guarantees. While reviewers acknowledged the importance of the problem, they raised a number of concerns that I believe the authors can try to address in the next version of the paper. For instance, reviewers questioned that the presented results are limited to only linear models while it is unclear how the proposed solution would fare for other models. Reviewers also suggested that the authors can better position their learning-augmented framework with related literatures (see reviewers' comments for the detailed list of related works).

**Additional Comments On Reviewer Discussion:**

Most of the reviewers' concerns remain after the rebuttals.

---

### Decision · Program_Chairs · 2025-01-22

Reject